# Zeroth-Order Stochastic Variance Reduction for Nonconvex Optimization

**Sijia Liu**[1]  **Bhavya Kailkhura**[2]  **Pin-Yu Chen**[1]  **Paishun Ting**[3]  **Shiyu Chang**[1]  **Lisa Amini**[1]

[1]MIT-IBM Watson AI Lab, IBM Research
[2]Lawrence Livermore National Laboratory
[3]University of Michigan, Ann Arbor

## Abstract

As application demands for zeroth-order (gradient-free) optimization accelerate, the need for variance reduced and faster converging approaches is also intensifying. This paper addresses these challenges by presenting: a) a comprehensive theoretical analysis of variance reduced zeroth-order (ZO) optimization, b) a novel variance reduced ZO algorithm, called ZO-SVRG, and c) an experimental evaluation of our approach in the context of two compelling applications, black-box chemical material classification and generation of adversarial examples from black-box deep neural network models. Our theoretical analysis uncovers an essential difficulty in the analysis of ZO-SVRG: the unbiased assumption on gradient estimates no longer holds. We prove that compared to its first-order counterpart, ZO-SVRG with a two-point random gradient estimator could suffer an additional error of order $O(1/b)$, where $b$ is the mini-batch size. To mitigate this error, we propose two accelerated versions of ZO-SVRG utilizing variance reduced gradient estimators, which achieve the best rate known for ZO stochastic optimization (in terms of iterations). Our extensive experimental results show that our approaches outperform other state-of-the-art ZO algorithms, and strike a balance between the convergence rate and the function query complexity.

## 1 Introduction

Zeroth-order (gradient-free) optimization is increasingly embraced for solving machine learning problems where explicit expressions of the gradients are difficult or infeasible to obtain. Recent examples have shown zeroth-order (ZO) based generation of prediction-evasive, black-box adversarial attacks on deep neural networks (DNNs) as effective as state-of-the-art white-box attacks, despite leveraging only the inputs and outputs of the targeted DNN [1–3]. Additional classes of applications include network control and management with time-varying constraints and limited computation capacity [4, 5], and parameter inference of black-box systems [6, 7]. ZO algorithms achieve gradient-free optimization by approximating the full gradient via gradient estimators based on only the function values [8, 9].

Although many ZO algorithms have recently been developed and analyzed [5, 10–18], they often suffer from the high variances of ZO gradient estimates, and in turn, hampered convergence rates. In addition, these algorithms are mainly designed for convex settings, which limits their applicability in a wide range of (non-convex) machine learning problems.

In this paper, we study the problem of design and analysis of variance reduced and faster converging nonconvex ZO optimization methods. To reduce the variance of ZO gradient estimates, one can draw motivations from similar ideas in the first-order regime. The stochastic variance reduced gradient (SVRG) is a commonly-used, effective first-order approach to reduce the variance [19–23]. Due to

the variance reduction, it improves the convergence rate of stochastic gradient descent (SGD) from $O(1/\sqrt{T})^1$ to $O(1/T)$, where $T$ is the total number of iterations.

Although SVRG has shown a great promise, applying similar ideas to ZO optimization is not a trivial task. The main challenge arises due to the fact that SVRG relies upon the assumption that a *stochastic* gradient is an *unbiased* estimate of the *true* batch/full gradient, which unfortunately does *not* hold in the ZO case. Therefore, it is an open question whether the ZO stochastic variance reduced gradient could enable faster convergence of ZO algorithms. In this paper, we attempt to fill the gap between ZO optimization and SVRG.

**Contributions** We propose and evaluate a novel ZO algorithm for nonconvex stochastic optimization, ZO-SVRG, which integrates SVRG with ZO gradient estimators. We show that compared to SVRG, ZO-SVRG achieves a similar convergence rate that decays linearly with $O(1/T)$ but up to an additional error correction term of order $1/b$, where $b$ is the mini-batch size. We show that this correction term will be eliminated as the full batch of data is used, corresponding to $b = n$ where $n$ is the number of data samples. In this scenario, ZO-SVRG would reduce to ZO gradient descent (ZO-GD) [13]. However, without a careful treatment, this correction term (e.g., when $b$ is small) could be a critical factor affecting the optimization performance. To mitigate this error term, we propose two accelerated ZO-SVRG variants, utilizing reduced variance gradient estimators. These yield a faster convergence rate towards $O(d/T)$, the best known iteration complexity bound for ZO stochastic optimization.

Our work offers a comprehensive study on how ZO gradient estimators affect SVRG on both iteration complexity (i.e., convergence rate) and function query complexity. Compared to the existing ZO algorithms, our methods can strike a balance between iteration complexity and function query complexity. To demonstrate the flexibility of our approach in managing this trade-off, we conduct an empirical evaluation of our proposed algorithms and other state-of-the-art algorithms on two diverse applications: black-box chemical material classification and generation of universal adversarial perturbations from black-box deep neural network models. Extensive experimental results and theoretical analysis validate the effectiveness of our approaches.

## 2    Related work

In ZO algorithms, a full gradient is typically approximated using either a one-point or a two-point gradient estimator, where the former acquires a gradient estimate $\hat{\nabla} f(\mathbf{x})$ by querying $f(\cdot)$ at a single random location close to $\mathbf{x}$ [10, 11], and the latter computes a finite difference using two random function queries [12, 13]. In this paper, we focus on the *two-point* gradient estimator since it has a lower variance and thus improves the complexity bounds of ZO algorithms.

Despite the meteoric rise of two-point based ZO algorithms, most of the work is restricted to convex problems [5, 14–18]. For example, a ZO mirror descent algorithm proposed by [14] has an exact rate $O(\sqrt{d}/\sqrt{T})$, where $d$ is the number of optimization variables. The same rate is obtained by bandit convex optimization [15] and ZO online alternating direction method of multipliers [5]. Current studies suggested that ZO algorithms typically agree with the iteration complexity of first-order algorithms up to a small-degree polynomial of the problem size $d$.

In contrast to the convex setting, non-convex ZO algorithms are comparatively under-studied except a few recent attempts [7, 13, 24–26]. Different from convex optimization, the stationary condition is used to measure the convergence of nonconvex methods. In [13], the ZO gradient descent (ZO-GD) algorithm was proposed for *deterministic* nonconvex programming, which yields $O(d/T)$ convergence rate. A stochastic version of ZO-GD (namely, ZO-SGD) studied in [24] achieves the rate of $O(\sqrt{d}/\sqrt{T})$. In [25], a ZO distributed algorithm was developed for multi-agent optimization, leading to $O(1/T + d/q)$ convergence rate. Here $q$ is the number of random directions used to construct a gradient estimate. In [7], an asynchronous ZO stochastic coordinate descent (ZO-SCD) was derived for parallel optimization and achieved the rate of $O(\sqrt{d}/\sqrt{T})$. In [26], a variant of ZO-SCD, known as ZO stochastic variance reduced coordinate (ZO-SVRC) descent, improved the convergence rate from $O(\sqrt{d}/\sqrt{T})$ to $O(d/T)$ under the same parameter setting for the gradient estimation. Although the authors in [26] considered the stochastic variance reduced technique, only a

coordinate descent algorithm using a coordinate-wise (deterministic) gradient estimator was studied. This motivates our study on a more general framework ZO-SVRG under different gradient estimators.

## 3 Preliminaries

Consider a nonconvex finite-sum problem of the form

$$\underset{\mathbf{x}\in\mathbb{R}^d}{\text{minimize}} \quad f(\mathbf{x}) := \frac{1}{n}\sum_{i=1}^{n} f_i(\mathbf{x}), \tag{1}$$

where $\{f_i(\mathbf{x})\}_{i=1}^{n}$ are $n$ individual nonconvex cost functions. The generic form (1) encompasses many machine learning problems, ranging from generalized linear models to neural networks. We next elaborate on assumptions of problem (1), and provide a background on ZO gradient estimators.

### 3.1 Assumptions

**A1:** Functions $\{f_i\}$ have $L$-Lipschitz continuous gradients ($L$-smooth), i.e., $\|\nabla f_i(\mathbf{x}) - \nabla f_i(\mathbf{y})\|_2 \leq L\|\mathbf{x} - \mathbf{y}\|_2$ for any $\mathbf{x}$ and $\mathbf{y}$, $i \in [n]$, and some $L < \infty$. Here $\|\cdot\|_2$ denotes the Euclidean norm, and for ease of notation $[n]$ represents the integer set $\{1, 2, \ldots, n\}$.

**A2:** The variance of stochastic gradients is bounded as $\frac{1}{n}\sum_{i=1}^{n} \|\nabla f_i(\mathbf{x}) - \nabla f(\mathbf{x})\|_2^2 \leq \sigma^2$. Here $\nabla f_i(\mathbf{x})$ can be viewed as a stochastic gradient of $\nabla f(\mathbf{x})$ by randomly picking an index $i \in [n]$.

Both A1 and A2 are the standard assumptions used in nonconvex optimization literature [7, 13, 23–26]. Note that A2 is milder than the assumption of bounded gradients [5, 25]. For example, if $\|\nabla f_i(\mathbf{x})\|_2 \leq \tilde{\sigma}$, then A2 is satisfied with $\sigma = 2\tilde{\sigma}$.

### 3.2 ZO gradient estimation

Given an individual cost function $f_i$ (or an arbitrary function under A1 and A2), a two-point random gradient estimator $\hat{\nabla} f_i(\mathbf{x})$ is defined by [13, 16]

$$\hat{\nabla} f_i(\mathbf{x}) = (d/\mu)\left[f_i(\mathbf{x} + \mu\mathbf{u}_i) - f_i(\mathbf{x})\right]\mathbf{u}_i, \text{ for } i \in [n], \tag{RandGradEst}$$

where recall that $d$ is the number of optimization variables, $\mu > 0$ is a smoothing parameter[2], and $\{\mathbf{u}_i\}$ are i.i.d. random directions drawn from a uniform distribution over a unit sphere [10, 15, 16]. In general, RandGradEst is a biased approximation to the true gradient $\nabla f_i(\mathbf{x})$, and its bias reduces as $\mu$ approaches zero. However, in a practical system, if $\mu$ is too small, then the function difference could be dominated by the system noise and fails to represent the function differential [7]. For $\mu > 0$, although the ZO gradient estimate introduces bias to the true gradient, it remains unbiased to the gradient of a so-called randomized smoothing function with parameter $\mu$; see Lemma 1 of Appendix A.1.

**Remark 1** *Instead of using a single sample $\mathbf{u}_i$ in RandGradEst, the average of q i.i.d. samples $\{\mathbf{u}_{i,j}\}_{j=1}^{q}$ can also be used for gradient estimation [5, 14, 25],*

$$\hat{\nabla} f_i(\mathbf{x}) = (d/(\mu q))\sum_{j=1}^{q}\left[f_i(\mathbf{x} + \mu\mathbf{u}_{i,j}) - f_i(\mathbf{x})\right]\mathbf{u}_{i,j}, \text{ for } i \in [n], \tag{Avg-RandGradEst}$$

*which we call an average random gradient estimator.*

In addition to RandGradEst and Avg-RandGradEst, the work [7, 26, 27] considered a coordinate-wise gradient estimator. Here every partial derivative is estimated via the two-point querying scheme under fixed direction vectors,

$$\hat{\nabla} f_i(\mathbf{x}) = \sum_{\ell=1}^{d} (1/(2\mu_\ell))\left[f_i(\mathbf{x} + \mu_\ell\mathbf{e}_\ell) - f_i(\mathbf{x} - \mu_\ell\mathbf{e}_\ell)\right]\mathbf{e}_l, \text{ for } i \in [n], \tag{CoordGradEst}$$

where $\mu_\ell > 0$ is a coordinate-wise smoothing parameter, and $\mathbf{e}_\ell \in \mathbb{R}^d$ is a standard basis vector with 1 at its $\ell$th coordinate, and 0s elsewhere. Compared to RandGradEst, CoordGradEst is *deterministic* and requires $d$ times more function queries. However, as will be evident later, it yields an improved iteration complexity (i.e., convergence rate). More details on ZO gradient estimation can be found in Appendix A.1.

| **Algorithm 1**: SVRG$(T, m, \{\eta_k\}, b, \tilde{\mathbf{x}}_0)$ | **Algorithm 2**: ZO-SVRG$(T, m, \{\eta_k\}, b, \tilde{\mathbf{x}}_0, \mu)$ |
|---|---|
| 1: **Input**: total number of iterations $T$, epoch length $m$, number of epochs $S = \lceil T/m \rceil$, step sizes $\{\eta_k\}_{k=0}^{m-1}$, mini-batch $b$, and $\tilde{\mathbf{x}}_0$. | 1: **Input**: In addition to parameters in SVRG, set smoothing parameter $\mu > 0$. |
| 2: **for** $s = 1, 2, \ldots, S$ **do** | 2: **for** $s = 1, 2, \ldots, S$ **do** |
| 3:     set $\mathbf{g}_s = \nabla f(\tilde{\mathbf{x}}_{s-1})$, $\mathbf{x}_0^s = \tilde{\mathbf{x}}_{s-1}$, | 3:     *compute ZO estimate* $\hat{\mathbf{g}}_s = \hat{\nabla} f(\tilde{\mathbf{x}}_{s-1})$, |
| 4:     **for** $k = 0, 1, \ldots, m-1$ **do** | 4:     set $\mathbf{x}_0^s = \tilde{\mathbf{x}}_{s-1}$, |
| 5:        choose mini-batch $\mathcal{I}_k$ of size $b$, | 5:     **for** $k = 0, 1, \ldots, m-1$ **do** |
| 6:        compute gradient blending via (2): $\mathbf{v}_k^s = \nabla f_{\mathcal{I}_k}(\mathbf{x}_k^s) - \nabla f_{\mathcal{I}_k}(\mathbf{x}_0^s) + \mathbf{g}_s$, | 6:        choose mini-batch $\mathcal{I}_k$ of size $b$, |
| 7:        update $\mathbf{x}_{k+1}^s = \mathbf{x}_k^s - \eta_k \mathbf{v}_k^s$, | 7:        *compute ZO gradient blending* (3): $\hat{\mathbf{v}}_k^s = \hat{\nabla} f_{\mathcal{I}_k}(\mathbf{x}_k^s) - \hat{\nabla} f_{\mathcal{I}_k}(\mathbf{x}_0^s) + \hat{\mathbf{g}}_s$, |
| 8:     **end for** | 8:        update $\mathbf{x}_{k+1}^s = \mathbf{x}_k^s - \eta_k \hat{\mathbf{v}}_k^s$, |
| 9:     set $\tilde{\mathbf{x}}_s = \mathbf{x}_m^s$, | 9:     **end for** |
| 10: **end for** | 10:     set $\tilde{\mathbf{x}}_s = \mathbf{x}_m^s$, |
| 11: **return** $\bar{\mathbf{x}}$ chosen uniformly random from $\{\{\mathbf{x}_k^s\}_{k=0}^{m-1}\}_{s=1}^S$. | 11: **end for** |
| | 12: **return** $\bar{\mathbf{x}}$ chosen uniformly random from $\{\{\mathbf{x}_k^s\}_{k=0}^{m-1}\}_{s=1}^S$. |

## 4 ZO stochastic variance reduced gradient (ZO-SVRG)

### 4.1 SVRG: from first-order to zeroth-order

It has been shown in [19, 20] that the first-order SVRG achieves the convergence rate $O(1/T)$, yielding $O(\sqrt{T})$ less iterations than the ordinary SGD for solving finite sum problems. The key step of SVRG[3] (Algorithm 1) is to generate an auxiliary sequence $\hat{\mathbf{x}}$ at which the full gradient is used as a reference in building a modified stochastic gradient estimate

$$\hat{\mathbf{g}} = \nabla f_{\mathcal{I}}(\mathbf{x}) - (\nabla f_{\mathcal{I}}(\hat{\mathbf{x}}) - \nabla f(\hat{\mathbf{x}})), \ \nabla f_{\mathcal{I}}(\mathbf{x}) = (1/b) \textstyle\sum_{i \in \mathcal{I}} \nabla f_i(\mathbf{x}) \tag{2}$$

where $\hat{\mathbf{g}}$ denotes the gradient estimate at $\mathbf{x}$, $\mathcal{I} \subseteq [n]$ is a mini-batch of size $b$ (chosen uniformly randomly[4]), and $\nabla f(\mathbf{x}) = \nabla f_{[n]}(\mathbf{x})$. The key property of (2) is that $\hat{\mathbf{g}}$ is an unbiased gradient estimate of $\nabla f(\mathbf{x})$. The gradient blending (2) is also motivated by a variance reduced technique known as control variate [28–30]. The link between SVRG and control variate is discussed in Appendix A.2.

In the ZO setting, the gradient blending (2) is approximated using only function values,

$$\hat{\mathbf{g}} = \hat{\nabla} f_{\mathcal{I}}(\mathbf{x}) - (\hat{\nabla} f_{\mathcal{I}}(\hat{\mathbf{x}}) - \hat{\nabla} f(\hat{\mathbf{x}})), \ \hat{\nabla} f_{\mathcal{I}}(\mathbf{x}) = (1/b) \textstyle\sum_{i \in \mathcal{I}} \hat{\nabla} f_i(\mathbf{x}), \tag{3}$$

where $\hat{\nabla} f(\mathbf{x}) = \hat{\nabla} f_{[n]}(\mathbf{x})$, and $\hat{\nabla} f_i$ is a ZO gradient estimate specified by RandGradEst, Avg-RandGradEst or CoordGradEst. Replacing (2) with (3) in SVRG (Algorithm 1) leads to a new ZO algorithm, which we call ZO-SVRG (Algorithm 2). We highlight that although ZO-SVRG is similar to SVRG except the use of ZO gradient estimators to estimate batch, mini-batch, as well as blended gradients, this seemingly minor difference yields an essential difficulty in the analysis of ZO-SVRG. That is, the unbiased assumption on gradient estimates used in SVRG no longer holds. Thus, a careful analysis of ZO-SVRG is much needed to ensure its optimization performance.

### 4.2 Convergence analysis

In what follows, we focus on the analysis of ZO-SVRG using RandGradEst. Later, we will study ZO-SVRG with Avg-RandGradEst and CoordGradEst. We start by investigating the second-order moment of the blended ZO gradient estimate $\hat{\mathbf{v}}_k^s$ in the form of (3); see Proposition 1.

**Proposition 1** *Suppose A2 holds and RandGradEst is used in Algorithm 2. The blended ZO gradient estimate $\hat{\mathbf{v}}_k^s$ in Step 7 of Algorithm 2 satisfies*

$$\mathbb{E}[\|\hat{\mathbf{v}}_k^s\|_2^2] \le \frac{4(b + 18\delta_n)d}{b} \mathbb{E}\left[\|\nabla f(\mathbf{x}_k^s)\|_2^2\right] + \frac{6(4d+1)L^2\delta_n}{b} \mathbb{E}\left[\|\mathbf{x}_k^s - \mathbf{x}_0^s\|_2^2\right] + \frac{(6\delta_n + b)L^2 d^2 \mu^2}{b} + \frac{72d\sigma^2\delta_n}{b}, \tag{4}$$

where $\delta_n = 1$ if the mini-batch contains i.i.d. samples from $[n]$ with replacement, and $\delta_n = I(b < n)$ if samples are randomly selected without replacement. Here $I(b < n)$ is $1$ if $b < n$, and $0$ if $b = n$.

**Proof**: See Appendix A.3. □

Compared to SVRG and its variants [20, 23], the error bound (4) involves a new error term $O(d\sigma^2/b)$ for $b < n$, which is induced by the second-order moment of RandGradEst (Appendix A.1). With the aid of Proposition 1, Theorem 1 provides the convergence rate of ZO-SVRG in terms of an upper bound on $\mathbb{E}[\|\nabla f(\bar{\mathbf{x}})\|^2]$ at the solution $\bar{\mathbf{x}}$.

**Theorem 1** *Suppose A1 and A2 hold, and the random gradient estimator* (RandGradEst) *is used. The output $\bar{\mathbf{x}}$ of Algorithm 2 satisfies*

$$\mathbb{E}\left[\|\nabla f(\bar{\mathbf{x}})\|_2^2\right] \leq \frac{f(\tilde{\mathbf{x}}_0) - f^*}{T\bar{\gamma}} + \frac{L\mu^2}{T\bar{\gamma}} + \frac{S\chi_m}{T\bar{\gamma}}, \tag{5}$$

*where $T = Sm$, $f^* = \min_{\mathbf{x}} f(\mathbf{x})$, $\bar{\gamma} = \min_{k \in [m]} \gamma_k$, $\chi_m = \sum_{k=0}^{m-1} \chi_k$, and*

$$\gamma_k = \frac{1}{2}\left(1 - \frac{c_{k+1}}{\beta_k}\right)\eta_k - \left(\frac{L}{2} + c_{k+1}\right)\frac{4db + 72d\delta_n}{b}\eta_k^2 \tag{6}$$

$$\chi_k = \left(1 - \frac{c_{k+1}}{\beta_k}\right)\frac{\mu^2 d^2 L^2}{4}\eta_k + \left(\frac{L}{2} + c_{k+1}\right)\frac{(6\delta_n + b)L^2 d^2 \mu^2 + 72d\sigma^2\delta_n}{b}\eta_k^2. \tag{7}$$

*In (6)-(7), $\beta_k$ is a positive parameter ensuring $\gamma_k > 0$, and the coefficients $\{c_k\}$ are given by*

$$c_k = \left[1 + \beta_k\eta_k + \frac{6(4d+1)L^2\delta_n\eta_k^2}{b}\right]c_{k+1} + \frac{3(4d+1)L^3\delta_n\eta_k^2}{b}, \quad c_m = 0. \tag{8}$$

**Proof**: See Appendix A.4. □

Compared to the convergence rate of SVRG as given in [20, Theorem 2], Theorem 1 exhibits two additional errors $(L\mu^2/(T\bar{\gamma}))$ and $(S\chi_m/(T\bar{\gamma}))$ due to the use of ZO gradient estimates. Roughly speaking, if we choose the smoothing parameter $\mu$ reasonably small, then the error $(L\mu^2/(T\bar{\gamma}))$ would reduce, leading to non-dominant effect on the convergence rate of ZO-SVRG. For the term $(S\chi_m/(T\bar{\gamma}))$, the quantity $\chi_m$ is more involved, relying on the epoch length $m$, the step size $\eta_k$, the smoothing parameter $\mu$, the mini-batch size $b$, and the number of optimization variables $d$. In order to acquire explicit dependence on these parameters and to explore deeper insights of convergence, we simplify (5) for a specific parameter setting, as formalized below.

**Corollary 1** *Suppose we set*

$$\mu = \frac{1}{\sqrt{dT}}, \quad \eta_k = \eta = \frac{\rho}{Ld}, \tag{9}$$

*$\beta_k = \beta = L$, and $m = \lceil \frac{d}{31\rho} \rceil$, where $0 < \rho \leq 1$ is a universal constant that is independent of $b$, $d$, $L$, and $T$. Then Theorem 1 implies $\frac{f(\tilde{\mathbf{x}}_0) - f^*}{T\bar{\gamma}} \leq O\left(\frac{d}{T}\right)$, $\frac{L\mu^2}{T\bar{\gamma}} \leq O\left(\frac{1}{T^2}\right)$, and $\frac{S\chi_m}{T\bar{\gamma}} \leq O\left(\frac{d}{T} + \frac{\delta_n}{b}\right)$, which yields*

$$\mathbb{E}\left[\|\nabla f(\bar{\mathbf{x}})\|_2^2\right] \leq O\left(\frac{d}{T} + \frac{\delta_n}{b}\right). \tag{10}$$

**Proof**: See Appendix A.5. □

It is worth mentioning that the condition on the value of smoothing parameter $\mu$ in Corollary 1 is less restrictive than several ZO algorithms[5]. For example, ZO-SGD in [24] required $\mu \leq O(d^{-1}T^{-1/2})$, and ZO-ADMM [5] and ZO-mirror descent [14] considered $\mu_t = O(d^{-1.5}t^{-1})$. Moreover, similar to [5], we set the step size $\eta$ linearly scaled with $1/d$. Compared to the aforementioned ZO algorithms [5, 14, 24], the convergence performance of ZO-SVRG in (10) has an improved (linear rather than sub-linear) dependence on $1/T$. However, it suffers an additional error of order $O(\delta_n/b)$ inherited from $(S\chi_m/(T\bar{\gamma}))$ in (5), which is also a consequence of the last error term in (4). We recall from the definition of $\delta_n$ in Proposition 1 that if $b < n$ or samples in the mini-batch are chosen independently from $[n]$, then $\delta_n = 1$. However, the error term is eliminated when $\mathcal{I}_k = [n]$ (corresponding to $\delta_n = 0$). In this case, ZO-SVRG (Algorithm 2) reduces to ZO-GD in [13] since Step 7 of Algorithm 2 becomes $\hat{\mathbf{v}}_k^s = \hat{\nabla} f(\mathbf{x}_k^s)$. A recent work [25, Theorem 1] also identified the possible side effect

$O(1/b)$ for $b < n$ in the context of ZO nonconvex multi-agent optimization using a method of multipliers. Note that a large mini-batch reduces the variance of RandGradEst and improves the convergence of ZO optimization methods. Although the tightness of the error bound (10) is not proven, we conjecture that the dependence on $T$ and $b$ could be optimal, since the form is consistent with SVRG, and the latter does not rely on the selected parameters in (9).

Lastly, we highlight that the theoretical analysis of ZO-SVRG is different from ZO-SVRC [26]. For the latter, the coordinate-wise (deterministic) gradient estimate is used and hence maintains Lipschitz continuity, which does not hold for a random gradient estimate. As a result, it becomes nontrivial to bound the distance of two random gradient estimates; see Appendix A.3. Moreover, reference [26] does not fully uncover the effect of dimension dependency on the convergence of ZO-SVRC. However, we clearly analyze this effect for ZO-SVRG in Corollary 1. Furthermore, our convergence analysis is performed under milder assumptions, while ZO-SVRC requires extra assumptions on gradients of coordinate-wise smoothing functions. In Sec. 6, we will compare the empirical performance of ZO-SVRC with our method through two real-life applications.

# 5 Acceleration of ZO-SVRG: Towards improved iteration complexity

In this section, we improve the iteration complexity of ZO-SVRG (Algorithm 2) by using Avg-RandGradEst and CoordGradEst, respectively. We start by comparing the squared errors of different gradient estimates to the true gradient $\nabla f$, as formalized in Proposition 2.

**Proposition 2** *Consider a gradient estimator $\hat{\nabla} f(\mathbf{x}) = \nabla f(\mathbf{x}) + \boldsymbol{\omega}$, then the squared error $\mathbb{E}[\|\boldsymbol{\omega}\|_2^2]$*

$$
\begin{cases}
\mathbb{E}\left[\|\boldsymbol{\omega}\|_2^2\right] \leq O\left(d\right)\|\nabla f(\mathbf{x})\|_2^2 + O\left(\mu^2 L^2 d^2\right) & \text{for RandGradEst,} \\
\mathbb{E}\left[\|\boldsymbol{\omega}\|_2^2\right] \leq O\left(\frac{q+d}{q}\right)\|\nabla f(\mathbf{x})\|_2^2 + O\left(\mu^2 L^2 d^2\right) & \text{for Avg-RandGradEst,} \\
\|\boldsymbol{\omega}\|_2^2 \leq O\left(L^2 d \sum_{\ell=1}^d \mu_\ell^2\right) & \text{for CoordGradEst.}
\end{cases}
\tag{11}
$$

**Proof**: See Appendix A.6. □

Proposition 2 shows that compared to CoordGradEst, RandGradEst and Avg-RandGradEst involve an additional error term within a factor $O(d)$ and $O((q+d)/q)$ of $\|\nabla f(\mathbf{x})\|_2^2$, respectively. Such an error is introduced by the second-order moment of gradient estimators using random direction samples [13, 14], and it decreases as the number of direction samples $q$ increases. On the other hand, all gradient estimators have a common error bounded by $O(\mu^2 L^2 d^2)$, where let $\mu_\ell = \mu$ for $\ell \in [d]$ in CoordGradEst. *If $\mu$ is specified as in (9), then we obtain the error term $O(d/T)$, consistent with the convergence rate of ZO-SVRG in Corollary 1.*

In Theorem 2, we show the effect of Avg-RandGradEst on the convergence rate of ZO-SVRG.

**Theorem 2** *Suppose A1 and A2 hold, and Avg-RandGradEst is used in Algorithm 2. Then $\mathbb{E}\left[\|\nabla f(\bar{\mathbf{x}})\|_2^2\right]$ is bounded in the same way as given in (5), where the parameters $\gamma_k$, $\chi_k$ and $c_k$ for $k \in [m]$ are modified by*

$$
\gamma_k = \frac{1}{2}\left(1 - \frac{c_{k+1}}{\beta_k}\right)\eta_k - \left(\frac{L}{2} + c_{k+1}\right)\frac{(72\delta_n + 4b)(q+d)}{bq}\eta_k^2,
$$

$$
\chi_k = \left(1 - \frac{c_{k+1}}{\beta_k}\right)\frac{\mu^2 d^2 L^2}{4}\eta_k + \left(\frac{L}{2} + c_{k+1}\right)\frac{(6\delta_n + b)(q+1)L^2 d^2 \mu^2 + 72(q+d)\sigma^2 \delta_n}{bq}\eta_k^2,
$$

$$
c_k = \left[1 + \beta_k \eta_k + \frac{6(4d+5q)L^2 \delta_n}{bq}\eta_k^2\right]c_{k+1} + \frac{3(4d+5q)L^3 \delta_n}{bq}\eta_k^2, \text{ with } c_m = 0.
$$

*Given the setting in Corollary 1 and $m = \lceil\frac{d}{55\rho}\rceil$, the convergence rate simplifies to*

$$
\mathbb{E}\left[\|\nabla f(\bar{\mathbf{x}})\|_2^2\right] \leq O\left(\frac{d}{T} + \frac{\delta_n}{b\min\{d,q\}}\right).
\tag{12}
$$

**Proof**: See Appendix A.7 □

By contrast with Corollary 1, it can be seen from (12) that the use of Avg-RandGradEst reduces the error $O(\delta_n/b)$ in (10) through multiple ($q$) direction samples. If $\frac{T}{db} \leq q \leq d$, then the convergence error under Ave-RandGradEst will be dominated by $O(d/T)$. Our empirical results show that a moderate choice of $q$ can significantly speed up the convergence of ZO-SVRG.

We next study the effect of the coordinate-wise gradient estimator (CoordGradEst) on the convergence rate of ZO-SVRG, as formalized in Theorem 3.

**Theorem 3** *Suppose A1 and A2 hold, and CoordGradEst with $\mu_\ell = \mu$ is used in Algorithm 2. Then*

$$\mathbb{E}\left[\|\nabla f(\bar{\mathbf{x}})\|_2^2\right] \leq \frac{f(\tilde{\mathbf{x}}_0) - f^*}{T\bar{\gamma}} + \frac{S\chi_m}{T\bar{\gamma}}, \tag{13}$$

*where $T$, $f^*$, $\bar{\gamma}$ and $\chi_m$ were defined in (5), the parameters $\gamma_k$, $\chi_k$ and $c_k$ for $k \in [m]$ are given by*

$$\gamma_k = \frac{1}{2}\left(1 - \frac{c_{k+1}}{\beta_k}\right)\eta_k - 4\left(\frac{L}{2} + c_{k+1}\right)\eta_k^2,$$

$$\chi_k = \left(\frac{1}{4} + \frac{c_{k+1}}{\beta_k}\right)\frac{L^2\mu^2 d^2}{2}\eta_k + \left(\frac{L}{2} + c_{k+1}\right)\mu^2 L^2 d^2 \eta_k^2,$$

$$c_k = \left(1 + \beta_k\eta_k + \frac{2dL^2\delta_n\eta_k^2}{b}\right)c_{k+1} + \frac{dL^3\delta_n\eta_k^2}{b}, \text{ with } c_m = 0,$$

*and $\beta_k$ is a positive parameter ensuring $\gamma_k > 0$. Given the specific setting in Corollary 1 and $m = \lceil\frac{d}{3\rho}\rceil$, the convergence rate simplifies to*

$$\mathbb{E}\left[\|\nabla f(\bar{\mathbf{x}})\|_2^2\right] \leq O\left(\frac{d}{T}\right). \tag{14}$$

**Proof**: See Appendix A.8. □

Theorem 3 shows that the use of CoordGradEst improves the iteration complexity, where the error of order $O(1/b)$ in Corollary 1 or $O(1/(b\min\{d,q\}))$ in Theorem 2 has been eliminated in (14). This improvement is benefited from the low variance of CoordGradEst shown by Proposition 2. We can also see this benefit by comparing $\chi_k$ in Theorem 3 with (7): the former avoids the term $(d\sigma^2/b)$. The disadvantage of CoordGradEst is the need of $d$ times more function queries than RandGradEst in gradient estimation.

Recall that RandGradEst, Avg-RandGradEst and CoordGradEst require $O(1)$, $O(q)$ and $O(d)$ function queries, respectively. In ZO-SVRG (Algorithm 2), the total number of gradient evaluations is given by $nS + bT$, where $T = mS$. Therefore, by fixing the number of iterations $T$, the function query complexity of ZO-SVRG using the studied estimators is then given by $O(nS + bT)$, $O(q(nS + bT))$ and $O(d(nS + bT))$, respectively. In Table 1, we summarize the convergence rates and the function query complexities of ZO-SVRG and its two variants, which we call ZO-SVRG-Ave and ZO-SVRG-Coord, respectively. For comparison, we also present the results of ZO-SGD [24] and ZO-SVRC [26], where the later updates $J$ coordinates per iteration within an epoch. Table 1 shows that ZO-SGD has the lowest query complexity but has the worst convergence rate. ZO-SVRG-coord yields the best convergence rate in the cost of high query complexity. By contrast, ZO-SVRG (with an appropriate mini-batch size) and ZO-SVRG-Ave could achieve better trade-offs between the convergence rate and the query complexity.

**Table 1:** Summary of convergence rate and function query complexity of our proposals given $T$ iterations.

| Method | Grad. estimator | Stepsize | Convergence rate (worst case as $b < n$) | Query complexity |
|---|---|---|---|---|
| ZO-SVRG | (RandGradEst) | $O\left(\frac{1}{d}\right)$ | $O\left(\frac{d}{T} + \frac{1}{b}\right)$ | $O\left(nS + bT\right)$ |
| ZO-SVRG-Ave | (Avg-RandGradEst) | $O(\frac{1}{d})$ | $O\left(\frac{d}{T} + \frac{1}{b\min\{d,q\}}\right)$ | $O\left(qnS + qbT\right)$ |
| ZO-SVRG-Coord | (CoordGradEst) | $O(\frac{1}{d})$ | $O(\frac{d}{T})$ | $O\left(dnS + dbT\right)$ |
| ZO-SGD [24] | (RandGradEst) | $O\left(\min\{\frac{1}{d}, \frac{1}{\sqrt{dT}}\}\right)$ | $O\left(\frac{\sqrt{d}}{\sqrt{T}}\right)$ | $O(bT)$ |
| ZO-SVRC [26] | (CoordGradEst) | $O\left(\frac{1}{n^\alpha}\right), \alpha \in (0,1)$ | $O\left(\frac{d}{T}\right)$ | $O\left(dnS + JbT\right)$ |

## 6 Applications and experiments

We evaluate the performance of our proposed algorithms on two applications: black-box classification and generating adversarial examples from black-box DNNs. The first application is motivated by a real-world material science problem, where a material is classified to either be a conductor or an insulator from a density function theory (DFT) based black-box simulator [31]. The second application arises in testing the robustness of a deployed DNN via iterative model queries [1, 3]. Since ZO-SVRG belongs to the class of ZO counterparts of first-order algorithms using random/deterministic gradient estimation, we compare it with ZO-SGD and ZO-SVRC, the most relevant methods to ZO-SVRG.

**Black-box binary classification**  We consider a non-linear least square problem [32, Sec. 3.2], i.e., problem (1) with $f_i(\mathbf{x}) = (y_i - \phi(\mathbf{x}; \mathbf{a}_i))^2$ for $i \in [n]$. Here $(\mathbf{a}_i, y_i)$ is the $i$th data sample containing feature vector $\mathbf{a}_i \in \mathbb{R}^d$ and label $y_i \in \{0, 1\}$, and $\phi(\mathbf{x}; \mathbf{a}_i)$ is a *black-box* function that only returns the function value given an input. The used dataset consists of $N = 1000$ crystalline materials/compounds extracted from Open Quantum Materials Database [33]. Each compound has $d = 145$ chemical features, and its label (0 is conductor and 1 is insulator) is determined by a DFT simulator [34]. Due to the black-box nature of DFT, the true $\phi$ is unknown[6]. We split the dataset into two equal parts, leading to $n = 500$ training samples and $(N - n)$ testing samples. We refer readers to Appendix A.10 for more details on our dataset and the setting of experiments.

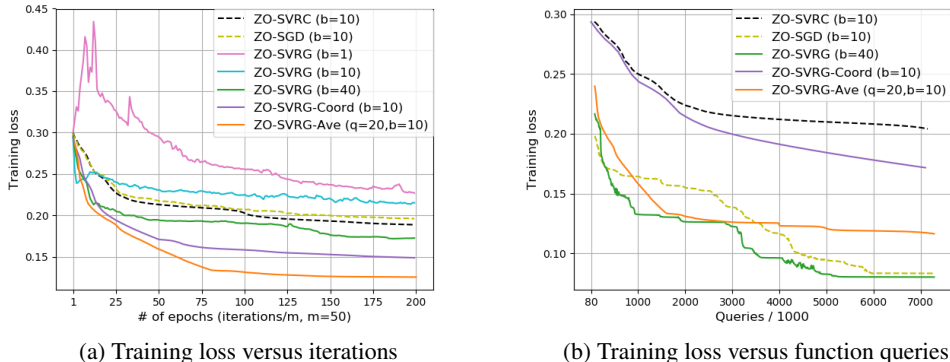

| (a) Training loss versus iterations | (b) Training loss versus function queries |

**Figure 2:** Comparison of different ZO algorithms for the task of chemical material classification.

**Table 2:** Testing error for chemical material classification using $7.3 \times 10^6$ function queries.

| Method | ZO-SGD [24] | ZO-SVRC [26] | ZO-SVRG | ZO-SVRG-Coord | ZO-SVRG-Ave |
|---|---|---|---|---|---|
| # of epochs | 14600 | 100 | 2920 | 50 | 365 |
| Error (%) | 12.56% | 23.70% | 11.18% | 20.67% | 15.26% |

In Fig. 2, we present the training loss against the number of epochs (i.e., iterations divided by the epoch length $m = 50$) and function queries. We compare our proposed algorithms ZO-SVRG, ZO-SVRG-Coord and ZO-SVRG-Ave with ZO-SGD [24] and ZO-SVRC [26]. Fig. 2-(a) presents the convergence trajectories of ZO algorithms as functions of the number of epochs, where ZO-SVRG is evaluated under different mini-batch sizes $b \in \{1, 10, 40\}$. We observe that the convergence error of ZO-SVRG decreases as $b$ increases, and for a small mini-batch size $b \leq 10$, ZO-SVRG likely converges to a neighborhood of a critical point as shown by Corollary 1. We also note that our proposed algorithms ZO-SVRG ($b = 40$), ZO-SVRG-Coord and ZO-SVRG-Ave have faster convergence speeds (i.e., less iteration complexity) than the existing algorithms ZO-SGD and ZO-SVRC. Particularly, the use of multiple random direction samples in Avg-RandGradEst significantly accelerates ZO-SVRG since the error of order $O(1/b)$ is reduced to $O(1/(bq))$ (see Table 1), leading to a non-dominant factor versus $O(d/T)$ in the convergence rate of ZO-SVRG-Ave. Fig. 2-(b) presents the training loss against the number of function queries. For the same experiment, Table 2 shows the number of iterations and the testing error of algorithms studied in Fig. 2-(b) using $7.3 \times 10^6$ function queries. We observe that the performance of CoordGradEst based algorithms (i.e., ZO-SVRC and ZO-SVRG-Coord) degrade due to the need of large number of function queries to construct coordinate-wise gradient estimates. By contrast, algorithms based on random gradient estimators (i.e., ZO-SGD, ZO-SVRG and ZO-SVRG-Ave) yield better both training and testing results, while ZO-SGD consumes an extremely large number of iterations (14600 epochs). As a result, ZO-SVRG ($b = 40$) and ZO-SVRG-Ave achieve better tradeoffs between the iteration and the function query complexity.

**Generation of adversarial examples from black-box DNNs**[7]  In image classification, adversarial examples refer to carefully crafted perturbations such that, when added to the natural images, are visually imperceptible but will lead the target model to misclassify. In the setting of 'zeroth order' attacks [2, 3, 35], the model parameters are hidden and acquiring its gradient is inadmissible. Only

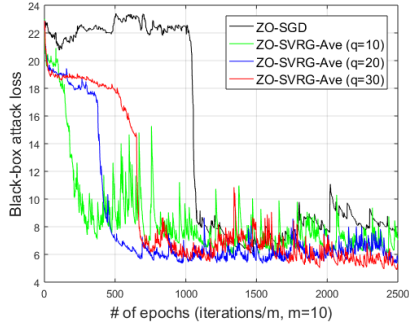

| Method | $\ell_2$ distortion |
|---|---|
| ZO-SGD | 5.22 |
| ZO-SVRG-Ave ($q = 10$) | 4.91 (6%) |
| ZO-SVRG-Ave ($q = 20$) | 3.91 (25%) |
| ZO-SVRG-Ave ($q = 30$) | 3.67 (30%) |

**Figure 3:** Comparison of ZO-SGD and ZO-SVRG-Ave for generation of universal adversarial perturbations from a black-box DNN. Left: Attack loss versus epochs. Right: $\ell_2$ distortion and improvement (%) with respect to ZO-SGD.

the model evaluations are accessible. We can then regard the task of generating a universal adversarial perturbation (to $n$ natural images) as an ZO optimization problem of the form (1). We elaborate on the problem formulation for generating adversarial examples in Appendix A.11.

We use a well-trained DNN[8] on the MNIST handwritten digit classification task as the target black-box model, which achieves 99.4% test accuracy on natural examples. Two ZO optimization methods, ZO-SGD and ZO-SVRG-Ave, are performed in our experiment. Note that ZO-SVRG-Ave reduces to ZO-SVRG when $q = 1$. We choose $n = 10$ images from the same class, and set the same parameters $b = 5$ and constant step size $30/d$ for both ZO methods, where $d = 28 \times 28$ is the image dimension. For ZO-SVRG-Ave, we set $m = 10$ and vary the number of random direction samples $q \in \{10, 20, 30\}$.

In Fig. 3, we show the black-box attack loss (against the number of epochs) as well as the least $\ell_2$ distortion of the successful (universal) adversarial perturbations. We observe that compared to ZO-SGD, ZO-SVRG-Ave offers a faster iteration convergence to a more accurate solution, and its convergence trajectory is more stable as $q$ becomes larger (due to the reduced variance of Avg-RandGradEst). Note that the sharp drop of attack loss in each method is caused by the hinge-like loss as part of the total loss function, which turns to 0 only if the attack becomes successful. In addition, ZO-SVRG-Ave improves the $\ell_2$ distortion of adversarial examples compared to ZO-SGD (e.g., 30% improvement when $q = 30$). We present the corresponding adversarial examples in Appendix A.11. In contrast with the iteration complexity, ZO-SVRG-Ave requires roughly $30\times$ ($q = 10$), $77\times$ ($q = 20$) and $380\times$ ($q = 30$) more function evaluations than ZO-SGD to reach a neighborhood of the smallest attack loss (e.g., 7 in our example). Furthermore, we present the black-box attack loss versus the number of query counts in Fig. A1 (Appendix A.11). As we can see, ZO-SVRG-Ave requires more queries than ZO-SGD to achieve the first significant drop in attack loss. However, by fixing the total number of queries ($10^7$), ZO-SVRG-Ave eventually converges to a lower loss than ZO-SGD: the former reaches the average loss 4.81 with std 0.32 (computed from the last 100 attack losses), while the latter reaches $6.74 \pm 0.46$.

## 7 Conclusion

In this paper, we studied ZO-SVRG, a new ZO nonconvex optimization method. We presented new convergence results beyond the existing work on ZO nonconvex optimization. We show that ZO-SVRG improves the convergence rate of ZO-SGD from $O(1/\sqrt{T})$ to $O(1/T)$ but suffers a new correction term of order $O(1/b)$. The is the side effect of combining a two-point random gradient estimators with SVRG. We then propose two accelerated variants of ZO-SVRG based on improved gradient estimators of reduced variances. We show an illuminating trade-off between the iteration and the function query complexity. Experimental results and theoretical analysis validate the effectiveness of our approaches compared to other state-of-the-art algorithms. In the future, we will compare ZO-SVRG with other derivative-free (non-gradient estimation based) methods for solving black-box optimization problems. It will also be interesting to study the problem of ZO distributed optimization, e.g., using CoordGradEst under a block coordinate descent framework [36].

**Acknowledgments**

This work was fully supported by the MIT-IBM Watson AI Lab. Bhavya Kailkhura was supported under the auspices of the U.S. Department of Energy by Lawrence Livermore National Laboratory under Contract DE-AC52-07NA27344 (LLNL-CONF-751658). The authors are also grateful to the anonymous reviewers for their helpful comments,

## Footnotes

[1]In the big $O$ notation, the constant numbers are ignored, and the dominant factors are kept.

[2]The parameter $\mu$ can be generalized to $\mu_i$ for $i \in [n]$. Here we assume $\mu_i = \mu$ for ease of representation.

[3]Different from the standard SVRG [19], we consider its mini-batch variant in [20].

[4]For mini-batch $\mathcal{I}$, SVRG [20] assumes i.i.d. samples with replacement, while a variant of SVRG (called SCSG) assumes samples without replacement [23]. This paper considers both sampling strategies.

[5]One exception is ZO-SCD [7] (and its variant ZO-SVRC [26]), where $\mu \leq O(1/\sqrt{T})$.

[6] One can mimic DFT simulator using a logistic function once the parameter $\mathbf{x}$ is learned from ZO algorithms.

[7] Code to reproduce experiments can be found at `https://github.com/IBM/ZOSVRG-BlackBox-Adv`

[8]`https://github.com/carlini/nn_robust_attacks`

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
