[Supplementary Material · Appendix_NIPS18_ZOSVRG_Final_Revise.pdf]

# Appendix

## A Supplementary material

### A.1 Zeroth-order (ZO) gradient estimators

With an abuse of notation, in this section let $f$ be an arbitrary function under assumptions A1 and A2. Lemma 1 shows the second-order statistics of RandGradEst.

**Lemma 1** *Suppose that Assumption A1 holds, and define a randomized smoothing function $f_\mu = \mathbb{E}_{\mathbf{u} \in U_b}[f(\mathbf{x} + \mu\mathbf{u})]$, where $U_b$ is a uniform distribution over the unit Euclidean ball. Then RandGradEst yields:*

*1) $f_\mu$ is L-smooth, and*

$$\nabla f_\mu(\mathbf{x}) = \mathbb{E}_{\mathbf{u}}\left[\hat{\nabla} f(\mathbf{x})\right], \tag{15}$$

*where $\mathbf{u}$ is drawn from the uniform distribution over the unit Euclidean sphere, and $\hat{\nabla} f(\mathbf{x})$ is given by RandGradEst.*

*2) For any $\mathbf{x} \in \mathbb{R}^d$,*

$$|f_\mu(\mathbf{x}) - f(\mathbf{x})| \leq \frac{L\mu^2}{2} \tag{16}$$

$$\|\nabla f_\mu(\mathbf{x}) - \nabla f(\mathbf{x})\|_2^2 \leq \frac{\mu^2 L^2 d^2}{4}, \tag{17}$$

$$\frac{1}{2}\|\nabla f(\mathbf{x})\|_2^2 - \frac{\mu^2 L^2 d^2}{4} \leq \|\nabla f_\mu(\mathbf{x})\|_2^2 \leq 2\|\nabla f(\mathbf{x})\|_2^2 + \frac{\mu^2 L^2 d^2}{2}. \tag{18}$$

*3) For any $\mathbf{x} \in \mathbb{R}^d$,*

$$\mathbb{E}_{\mathbf{u}}\left[\|\hat{\nabla} f(\mathbf{x}) - \nabla f_\mu(\mathbf{x})\|_2^2\right] \leq \mathbb{E}_{\mathbf{u}}\left[\|\hat{\nabla} f(\mathbf{x})\|_2^2\right] \leq 2d\|\nabla f(\mathbf{x})\|_2^2 + \frac{\mu^2 L^2 d^2}{2}. \tag{19}$$

**Proof:** First, by using [16, Lemma 4.1.a] (also see [37] and [13]), we immediately obtain that $f_\mu$ is $L_\mu$ smooth with $L_\mu \leq L$, and

$$\nabla f_\mu(\mathbf{x}) = \mathbb{E}_{\mathbf{u}}\left[\frac{d}{\mu} f(\mathbf{x} + \mu\mathbf{u})\mathbf{u}\right]. \tag{20}$$

Since $\mathbb{E}_{\mathbf{u}}[(d/\mu)f(\mathbf{x})\mathbf{u}] = 0$, we obtain (15).

Next, we obtain (16)-(18) based on [16, Lemma 4.1.b]. Moreover, we have

$$\begin{aligned}
\|\nabla f_\mu(\mathbf{x})\|_2^2 &= \|\nabla f_\mu(\mathbf{x}) - \nabla f(\mathbf{x}) + \nabla f(\mathbf{x})\|_2^2 \\
&\leq 2\|\nabla f(\mathbf{x})\|_2^2 + 2\|\nabla f_\mu(\mathbf{x}) - \nabla f(\mathbf{x})\|_2^2 \\
&\overset{(17)}{\leq} 2\|\nabla f(\mathbf{x})\|_2^2 + \frac{\mu^2 d^2 L^2}{2},
\end{aligned} \tag{21}$$

where the first inequality holds due to Lemma 6. Similarly, we have

$$\|\nabla f(\mathbf{x})\|_2^2 = \|\nabla f_\mu(\mathbf{x}) + \nabla f(\mathbf{x}) - \nabla f_\mu(\mathbf{x})\|_2^2 \overset{(17)}{\leq} 2\|\nabla f_\mu(\mathbf{x})\|_2^2 + \frac{\mu^2 d^2 L^2}{2}, \tag{22}$$

which yields

$$\|\nabla f_\mu(\mathbf{x})\|_2^2 \geq \frac{1}{2}\|\nabla f(\mathbf{x})\|_2^2 - \frac{\mu^2 L^2 d^2}{4}. \tag{23}$$

In (19), the first inequality holds due to (15) and $\mathbb{E}[\|\mathbf{a} - \mathbb{E}[\mathbf{a}]\|_2^2] \leq \mathbb{E}[\|\mathbf{a}\|_2^2]$ for a random variable $\mathbf{a}$. And the second inequality of (19) holds due to [16, Lemma 4.1.b]. The proof is now complete. □

In Lemma 2, we show the properties of Avg-RandGradEst.

**Lemma 2** *Following the conditions of Lemma 1, then Avg-RandGradEst yields:*

*1) For any $\mathbf{x} \in \mathbb{R}^d$*

$$\nabla f_\mu(\mathbf{x}) = \mathbb{E}\left[\hat{\nabla} f(\mathbf{x})\right], \tag{24}$$

*where $\hat{\nabla} f(\mathbf{x})$ is given by Avg-RandGradEst.*

*2) For any $\mathbf{x} \in \mathbb{R}^d$*

$$\mathbb{E}\left[\|\hat{\nabla} f(\mathbf{x}) - \nabla f_\mu(\mathbf{x})\|_2^2\right] \leq \mathbb{E}\left[\|\hat{\nabla} f(\mathbf{x})\|_2^2\right] \leq 2\left(1 + \frac{d}{q}\right)\|\nabla f(\mathbf{x})\|_2^2 + \left(1 + \frac{1}{q}\right)\frac{\mu^2 L^2 d^2}{2}. \tag{25}$$

**Proof**: Since $\{\mathbf{u}_i\}_{i=1}^q$ are i.i.d. random vectors, we have

$$\mathbb{E}\left[\hat{\nabla} f(\mathbf{x})\right] = \mathbb{E}_{\mathbf{u}_i}\left[\hat{\nabla} f(\mathbf{x}; \mathbf{u}_i)\right] \overset{(15)}{=} \nabla f_\mu(\mathbf{x}), \tag{26}$$

where $\hat{\nabla} f(\mathbf{x}; \mathbf{u}_i) := \frac{d}{\mu}[f(\mathbf{x} + \mu\mathbf{u}_i) - f(\mathbf{x})]\mathbf{u}_i$.

In (25), the first inequality holds due to (24) and $\mathbb{E}[\|\mathbf{a} - \mathbb{E}[\mathbf{a}]\|_2^2] \leq \mathbb{E}[\|\mathbf{a}\|_2^2]$ for a random variable $\mathbf{a}$. Next, we bound the second moment of $\hat{\nabla} f(\mathbf{x})$

$$\mathbb{E}\left[\left\|\hat{\nabla} f(\mathbf{x})\right\|_2^2\right] = \mathbb{E}\left[\left\|\frac{1}{q}\sum_{i=1}^q \left(\hat{\nabla} f(\mathbf{x}; \mathbf{u}_i) - \nabla f_\mu(\mathbf{x})\right) + \nabla f_\mu(\mathbf{x})\right\|_2^2\right]$$

$$= \|\nabla f_\mu(\mathbf{x})\|_2^2 + \mathbb{E}\left[\left\|\frac{1}{q}\sum_{i=1}^q \left(\hat{\nabla} f(\mathbf{x}; \mathbf{u}_i) - \nabla f_\mu(\mathbf{x})\right)\right\|_2^2\right]$$

$$= \|\nabla f_\mu(\mathbf{x})\|_2^2 + \frac{1}{q}\mathbb{E}\left[\left\|\hat{\nabla} f(\mathbf{x}; \mathbf{u}_1) - \nabla f_\mu(\mathbf{x})\right\|_2^2\right], \tag{27}$$

where the expectation is taken with respect to i.i.d. random vectors $\{\mathbf{u}_i\}$, and we have used the fact that $\mathbb{E}[\|\hat{\nabla} f(\mathbf{x}; \mathbf{u}_i) - \nabla f_\mu(\mathbf{x})\|_2^2] = \mathbb{E}[\|\hat{\nabla} f(\mathbf{x}; \mathbf{u}_1) - \nabla f_\mu(\mathbf{x})\|_2^2]$ for any $i$. Substituting (18) and (19) into (27), we obtain (25). $\square$

In Lemma 3, we demonstrate the properties of CoordGradEst.

**Lemma 3** *Let Assumption A1 hold and define $f_{\mu_\ell}(\mathbf{x}) = \mathbb{E}_{u \sim U[-\mu_\ell, \mu_\ell]} f(\mathbf{x} + u\mathbf{e}_\ell)$, where $U[-\mu_\ell, \mu_\ell]$ denotes the uniform distribution at the interval $[-\mu_\ell, \mu_\ell]$. We then have:*

*1) $f_{\mu_\ell}$ is L-smooth, and*

$$\hat{\nabla} f(\mathbf{x}) = \sum_{\ell=1}^d \frac{\partial f_{\mu_\ell}(\mathbf{x})}{\partial x_\ell}\mathbf{e}_\ell, \tag{28}$$

*where $\hat{\nabla} f(\mathbf{x})$ is defined by CoordGradEst, and $\partial f/\partial x_\ell$ denotes the partial derivative with respect to the $\ell$th coordinate.*

*2) For $\ell \in [d]$,*

$$|f_{\mu_\ell}(\mathbf{x}) - f(\mathbf{x})| \leq \frac{L\mu_\ell^2}{2}, \tag{29}$$

$$\left|\frac{\partial f_{\mu_\ell}(\mathbf{x})}{\partial x_\ell} - \frac{\partial f(\mathbf{x})}{\partial x_\ell}\right| \leq \frac{L\mu_\ell}{2}. \tag{30}$$

*3) For $\ell \in [d]$,*

$$\left\|\hat{\nabla} f(\mathbf{x}) - \nabla f(\mathbf{x})\right\|_2^2 \leq \frac{L^2 d}{4}\sum_{\ell=1}^d \mu_\ell^2. \tag{31}$$

**Proof**: For the $\ell$th coordinate, it is known from [7, Lemma 6] that $f_{\mu_\ell}$ is $L$-smooth and

$$\frac{\partial f_{\mu_\ell}(\mathbf{x})}{\partial x_\ell} = \frac{f(\mathbf{x} + \mu_\ell \mathbf{e}_\ell) - f(\mathbf{x} - \mu_\ell \mathbf{e}_\ell)}{2\mu_\ell}. \tag{32}$$

Based on (32) and the definition of CoordGradEst, we then obtain (28).

The inequalities (29) and (30) have been proved by [7, Lemma 6].

Based on (28) and (30), we have

$$\left\| \hat{\nabla} f(\mathbf{x}) - \nabla f(\mathbf{x}) \right\|_2^2 \overset{(28)}{=} \left\| \sum_{\ell=1}^{d} \left( \frac{\partial f_{\mu_\ell}(\mathbf{x})}{\partial x_\ell} - \frac{\partial f(\mathbf{x})}{\partial x_\ell} \right) \mathbf{e}_\ell \right\|_2^2$$

$$\leq d \sum_{\ell=1}^{d} \left\| \frac{\partial f_{\mu_\ell}(\mathbf{x})}{\partial x_\ell} - \frac{\partial f(\mathbf{x})}{\partial x_\ell} \right\|_2^2 \leq \frac{L^2 d}{4} \sum_{\ell=1}^{d} \mu_\ell^2,$$

where the first inequality holds due to Lemma 6 in Sec. A.9. The proof is now complete. $\quad\square$

## A.2 Control variates

The gradient blending in Step 6 of SVRG (Algorithm 1) can be interpreted using control variate [28–30]. If we view $\hat{\mathbf{g}}_0 := \nabla f_{\mathcal{I}}(\mathbf{x})$ as the raw gradient estimate at $\mathbf{x}$, and $\mathbf{c} := \nabla f_{\mathcal{I}}(\hat{\mathbf{x}})$ as a control variate satisfying $\mathbb{E}[\mathbf{c}] = \nabla f(\hat{\mathbf{x}})$, then the gradient blending (2) becomes a gradient estimate modified by a control variate, $\hat{\mathbf{g}} = \hat{\mathbf{g}}_0 - (\mathbf{c} - \mathbb{E}[\mathbf{c}])$. Here $\hat{\mathbf{g}}$ has the same expectation as $\hat{\mathbf{g}}_0$, i.e., $\mathbb{E}[\hat{\mathbf{g}}] = \mathbb{E}[\hat{\mathbf{g}}_0] = \nabla f(\mathbf{x})$, however, has a lower variance when $\mathbf{c}$ is positively correlated with $\mathbf{g}_0$ (see a detailed analysis as below).

Consider the following gradient estimator,

$$\hat{\mathbf{g}} = \hat{\mathbf{g}}_0 - \eta(\mathbf{c} - \mathbb{E}[\mathbf{c}]), \tag{33}$$

where $\hat{\mathbf{g}}_0$ is a given (raw) gradient estimate, $\eta$ is an unknown coefficient, and $\mathbf{c}$ is a control variate. It is clear that $\hat{\mathbf{g}}$ has the same expectation as $\hat{\mathbf{g}}_0$. We then study the effect of $\mathbf{c}$ on the variance of $\hat{\mathbf{g}}$,

$$\mathrm{tr}(\mathrm{cov}(\hat{\mathbf{g}})) = \mathrm{tr}(\mathrm{cov}(\hat{\mathbf{g}}_0)) + \eta^2 \, \mathrm{tr}(\mathrm{cov}(\mathbf{c})) - 2\eta \, \mathrm{tr}(\mathrm{cov}(\hat{\mathbf{g}}_0, \mathbf{c})), \tag{34}$$

where $\mathrm{tr}(\cdot)$ denotes the trace operator, and $\mathrm{cov}(\cdot)$ is the covariance operator. When $\eta = \frac{\mathrm{tr}(\mathrm{cov}(\hat{\mathbf{g}}_0, \mathbf{c}))}{\mathrm{tr}(\mathrm{cov}(\mathbf{c}))}$, the variance of $\hat{\mathbf{g}}$ in (34) is then minimized, leading to

$$\mathrm{tr}(\mathrm{cov}(\hat{\mathbf{g}})) = \mathrm{tr}(\mathrm{cov}(\hat{\mathbf{g}}_0)) \left( 1 - \rho(\hat{\mathbf{g}}_0, \mathbf{c})^2 \right), \tag{35}$$

where $\rho(\hat{\mathbf{g}}_0, \mathbf{c}) = \frac{\mathrm{tr}(\mathrm{cov}(\hat{\mathbf{g}}_0, \mathbf{c}))}{\sqrt{\mathrm{tr}(\mathrm{cov}(\hat{\mathbf{g}}_0))}\sqrt{\mathrm{tr}(\mathrm{cov}(\mathbf{c}))}}$. In (35), $\rho(\hat{\mathbf{g}}_0, \mathbf{c})$ indicates the correlation strength between $\hat{\mathbf{g}}_0$ and $\mathbf{c}$. Therefore, the gradient estimate $\hat{\mathbf{g}}$ has a smaller variance than $\hat{\mathbf{g}}_0$ when the control variate $\mathbf{c}$ is *positively* correlated with the latter. Moreover, if $\mathbf{c}$ is chosen similar to $\hat{\mathbf{g}}$, then $\eta$ would be close to 1.

## A.3 Proof of Proposition 1

In Algorithm 2, we recall that the mini-batch $\mathcal{I}$ is chosen uniformly randomly. It is known from Lemma 4 and Lemma 5 that

$$\mathbb{E}_{\mathcal{I}_k}[\hat{\nabla} f_{\mathcal{I}_k}(\mathbf{x}_k^s) - \hat{\nabla} f_{\mathcal{I}_k}(\mathbf{x}_0^s)] = \hat{\nabla} f(\mathbf{x}_k^s) - \hat{\nabla} f(\mathbf{x}_0^s). \tag{36}$$

We then rewrite $\hat{\mathbf{v}}_k^s$ as

$$\hat{\mathbf{v}}_k^s = \hat{\nabla} f_{\mathcal{I}_k}(\mathbf{x}_k^s) - \hat{\nabla} f_{\mathcal{I}_k}(\mathbf{x}_0^s) - \mathbb{E}_{\mathcal{I}_k}[\hat{\nabla} f_{\mathcal{I}_k}(\mathbf{x}_k^s) - \hat{\nabla} f_{\mathcal{I}_k}(\mathbf{x}_0^s)] + \hat{\nabla} f(\mathbf{x}_k^s). \tag{37}$$

Taking the expectation of $\|\mathbf{v}_k^s\|_2^2$ with repsect to all the random variables, we have

$$\mathbb{E}\left[\|\hat{\mathbf{v}}_k^s\|_2^2\right] \leq 2\mathbb{E}\left[\|\hat{\nabla} f_{\mathcal{I}_k}(\mathbf{x}_k^s) - \hat{\nabla} f_{\mathcal{I}_k}(\mathbf{x}_0^s) - \mathbb{E}_{\mathcal{I}_k}[\hat{\nabla} f_{\mathcal{I}_k}(\mathbf{x}_k^s) - \hat{\nabla} f_{\mathcal{I}_k}(\mathbf{x}_0^s)]\|_2^2\right] + 2\mathbb{E}\left[\|\hat{\nabla} f(\mathbf{x}_k^s)\|_2^2\right]$$

$$\leq 2\mathbb{E}\left[\|\hat{\nabla} f_{\mathcal{I}_k}(\mathbf{x}_k^s) - \hat{\nabla} f_{\mathcal{I}_k}(\mathbf{x}_0^s) - \mathbb{E}_{\mathcal{I}_k}[\hat{\nabla} f_{\mathcal{I}_k}(\mathbf{x}_k^s) - \hat{\nabla} f_{\mathcal{I}_k}(\mathbf{x}_0^s)]\|_2^2\right]$$

$$+ 4d\mathbb{E}\left[\|\nabla f(\mathbf{x}_k^s)\|_2^2\right] + \mu^2 d^2 L^2, \tag{38}$$

where the first inequality holds due to Lemma 6, and the second inequality holds due to (19). Based on (36), we note that the following holds

$$\sum_{i=1}^{n}\left\{\hat{\nabla} f_i(\mathbf{x}_k^s) - \hat{\nabla} f_i(\mathbf{x}_k^s) - \mathbb{E}_{\mathcal{I}_k}[\hat{\nabla} f_{\mathcal{I}_k}(\mathbf{x}_k^s) - \hat{\nabla} f_{\mathcal{I}_k}(\mathbf{x}_0^s)]\right\}$$
$$=n(\hat{\nabla} f(\mathbf{x}_k^s) - \hat{\nabla} f(\mathbf{x}_0^s)) - n(\hat{\nabla} f(\mathbf{x}_k^s) - \hat{\nabla} f(\mathbf{x}_0^s)) = \mathbf{0}. \tag{39}$$

Based on (39) and applying Lemma 4 and 5, the first term at the right hand side (RHS) of (38) yields

$$\mathbb{E}\left[\|\hat{\nabla} f_{\mathcal{I}_k}(\mathbf{x}_k^s) - \hat{\nabla} f_{\mathcal{I}_k}(\mathbf{x}_0^s) - \mathbb{E}_{\mathcal{I}_k}[\hat{\nabla} f_{\mathcal{I}_k}(\mathbf{x}_k^s) - \hat{\nabla} f_{\mathcal{I}_k}(\mathbf{x}_0^s)]\|_2^2\right]$$

$$\leq \frac{\delta_n}{bn}\sum_{i=1}^{n}\mathbb{E}\left[\|\hat{\nabla} f_i(\mathbf{x}_k^s) - \hat{\nabla} f_i(\mathbf{x}_0^s) - (\hat{\nabla} f(\mathbf{x}_k^s) - \hat{\nabla} f(\mathbf{x}_0^s))\|_2^2\right]$$

$$=\mathbb{E}\left[\frac{\delta_n}{b}\left(\frac{1}{n}\sum_{i=1}^{n}\|\hat{\nabla} f_i(\mathbf{x}_k^s) - \hat{\nabla} f_i(\mathbf{x}_0^s)\|_2^2 - \|\hat{\nabla} f(\mathbf{x}_k^s) - \hat{\nabla} f(\mathbf{x}_0^s)\|_2^2\right)\right]$$

$$\leq \frac{\delta_n}{bn}\sum_{i=1}^{n}\mathbb{E}\left[\|\hat{\nabla} f_i(\mathbf{x}_k^s) - \hat{\nabla} f_i(\mathbf{x}_0^s)\|_2^2\right]. \tag{40}$$

where the first inequality holds due to Lemma 4 and 5 (taking the expectation with respect to mini-batch $\mathcal{I}$), we define $\delta_n$ as

$$\delta_n = \begin{cases} 1 & \text{if } \mathcal{I} \text{ contains i.i.d. samples with replacement (Lemma 4)} \\ I(b < n) & \text{if } \mathcal{I} \text{ contains samples without replacement (Lemma 5),} \end{cases} \tag{41}$$

$I(b < n) = 1$ if $b < n$ and 0 otherwise, and the second equality in (40) holds since $\frac{1}{n}\sum_{i=1}^{n}\|\mathbf{x}_i - \mathbf{a}\|_2^2 = \frac{1}{n}\sum_{i=1}^{n}\|\mathbf{x}_i\|_2^2 - \|\mathbf{a}\|_2^2$ when $\mathbf{a} = \frac{1}{n}\sum_{i=1}^{n}\mathbf{x}_i$.

Substituting (40) into (38), we obtain

$$\mathbb{E}\left[\|\hat{\mathbf{v}}_k^s\|_2^2\right] \leq \frac{2\delta_n}{bn}\sum_{i=1}^{n}\mathbb{E}\left[\|\hat{\nabla} f_i(\mathbf{x}_k^s) - \hat{\nabla} f_i(\mathbf{x}_0^s)\|_2^2\right] + 4d\mathbb{E}\left[\|\nabla f(\mathbf{x}_k^s)\|_2^2\right] + \mu^2 d^2 L^2. \tag{42}$$

Similar to Lemma 1, we introduce a smoothing function $f_{i,\mu}$ of $f_i$, and continue to bound the first term at the right hand side (RHS) of (42). This yields

$$\mathbb{E}\left[\|\hat{\nabla} f_i(\mathbf{x}_k^s) - \hat{\nabla} f_i(\mathbf{x}_0^s)\|_2^2\right]$$
$$\overset{(130)}{\leq} 3\mathbb{E}\left[\|\hat{\nabla} f_i(\mathbf{x}_k^s) - \nabla f_{i,\mu}(\mathbf{x}_k^s)\|_2^2\right] + 3\mathbb{E}\left[\|\hat{\nabla} f_{i,\mu}(\mathbf{x}_0^s) - \nabla f_i(\mathbf{x}_0^s)\|_2^2\right]$$
$$\quad + 3\mathbb{E}\left[\|\nabla f_{i,\mu}(\mathbf{x}_k^s) - \nabla f_{i,\mu}(\mathbf{x}_0^s)\|_2^2\right]$$
$$\overset{(19)}{\leq} 6d\mathbb{E}[\|\nabla f_i(\mathbf{x}_k^s)\|_2^2] + 6d\mathbb{E}[\|\nabla f_i(\mathbf{x}_0^s)\|_2^2] + 3L^2 d^2 \mu^2 + 3\mathbb{E}\left[\|\nabla f_{i,\mu}(\mathbf{x}_k^s) - \nabla f_{i,\mu}(\mathbf{x}_0^s)\|_2^2\right]. \tag{43}$$

Since both $f_i$ and $f_{i,\mu}$ are $L$-smooth (A1 and Lemma 1), we have

$$\mathbb{E}\left[\|\nabla f_{i,\mu}(\mathbf{x}_k^s) - \nabla f_{i,\mu}(\mathbf{x}_0^s)\|_2^2\right] \leq L^2 \mathbb{E}\left[\|\mathbf{x}_k^s - \mathbf{x}_0^s\|_2^2\right], \tag{44}$$

$$\mathbb{E}\left[\|\nabla f_i(\mathbf{x}_0^s)\|_2^2\right] \leq 2\mathbb{E}\left[\|\nabla f_i(\mathbf{x}_0^s) - \nabla f_i(\mathbf{x}_k^s)\|_2^2\right] + 2\mathbb{E}\left[\|\nabla f_i(\mathbf{x}_k^s)\|_2^2\right]$$
$$\leq 2L^2 \mathbb{E}\left[\|\mathbf{x}_0^s - \mathbf{x}_k^s\|_2^2\right] + 2\mathbb{E}\left[\|\nabla f_i(\mathbf{x}_k^s)\|_2^2\right]. \tag{45}$$

Substituting (44) and (45) into (43), we obtain

$$\mathbb{E}\left[\|\hat{\nabla} f_i(\mathbf{x}_k^s) - \hat{\nabla} f_i(\mathbf{x}_0^s)\|_2^2\right]$$
$$\leq 18d\mathbb{E}[\|\nabla f_i(\mathbf{x}_k^s)\|_2^2] + (12d + 3)L^2 \mathbb{E}\left[\|\mathbf{x}_0^s - \mathbf{x}_k^s\|_2^2\right] + 3L^2 d^2 \mu^2$$
$$\leq 36d\mathbb{E}\left[\|\nabla f_i(\mathbf{x}_k^s) - \nabla f(\mathbf{x}_k^s)\|_2^2\right] + 36d\mathbb{E}\left[\|\nabla f(\mathbf{x}_k^s)\|_2^2\right]$$
$$\quad + (12d + 3)L^2 \mathbb{E}\left[\|\mathbf{x}_0^s - \mathbf{x}_k^s\|_2^2\right] + 3L^2 d^2 \mu^2. \tag{46}$$

Furthermore, we obtain

$$\frac{1}{n}\sum_{i=1}^{n}\mathbb{E}\left[\|\hat{\nabla}f_i(\mathbf{x}_k^s) - \hat{\nabla}f_i(\mathbf{x}_0^s)\|_2^2\right]$$

$$\leq 36d\mathbb{E}\left[\frac{1}{n}\sum_{i=1}^{n}\|\nabla f_i(\mathbf{x}_k^s) - \nabla f(\mathbf{x}_k^s)\|_2^2\right] + 36d\mathbb{E}\left[\|\nabla f(\mathbf{x}_k^s)\|_2^2\right]$$

$$+ (12d+3)L^2\mathbb{E}\left[\|\mathbf{x}_0^s - \mathbf{x}_k^s\|_2^2\right] + 3L^2d^2\mu^2 \tag{47}$$

$$\leq 36d\sigma^2 + 36d\mathbb{E}\left[\|\nabla f(\mathbf{x}_k^s)\|_2^2\right] + (12d+3)L^2\mathbb{E}\left[\|\mathbf{x}_0^s - \mathbf{x}_k^s\|_2^2\right] + 3L^2d^2\mu^2, \tag{48}$$

where the last inequality holds due to Assumption A2.

Substituting (47) into (42), we have

$$\mathbb{E}\left[\|\hat{\mathbf{v}}_k^s\|_2^2\right] \leq \frac{6\delta_n(4d+1)L^2}{b}\mathbb{E}\left[\|\mathbf{x}_0^s - \mathbf{x}_k^s\|_2^2\right]$$

$$+ \left(4d + \frac{72d\delta_n}{b}\right)\mathbb{E}\left[\|\nabla f(\mathbf{x}_k^s)\|_2^2\right] + \left(1 + \frac{6\delta_n}{b}\right)d^2L^2\mu^2 + \frac{72d\sigma^2\delta_n}{b}. \tag{49}$$

The proof is now complete. $\qquad\square$

## A.4 Proof of Theorem 1

Since $f_\mu$ is $L$-smooth (Lemma 1), from Lemma 7 in Sec. A.9 we have

$$f_\mu(\mathbf{x}_{k+1}^s) \leq f_\mu(\mathbf{x}_k^s) + \langle\nabla f_\mu(\mathbf{x}_k^s), \mathbf{x}_{k+1}^s - \mathbf{x}_k^s\rangle + \frac{L}{2}\|x_{k+1}^s - \mathbf{x}_k^s\|_2^2$$

$$= f_\mu(\mathbf{x}_k^s) - \eta_k\langle\nabla f_\mu(\mathbf{x}_k^s), \hat{\mathbf{v}}_k^s\rangle + \frac{L}{2}\eta_k^2\|\hat{\mathbf{v}}_k^s\|_2^2, \tag{50}$$

where the last equality holds due to $\mathbf{x}_{k+1}^s = \mathbf{x}_k^s - \eta_k\hat{\mathbf{v}}_k^s$. Since $\mathbf{x}_k^s$ and $\mathbf{x}_0^s$ are independent of $\mathcal{I}_k$ and random directions $\mathbf{u}$ used for ZO gradient estimates, from (15) we obtain

$$\mathbb{E}_{\mathbf{u},\mathcal{I}_k}\left[\hat{\mathbf{v}}_k^s\right] = \mathbb{E}_{\mathbf{u},\mathcal{I}_k}\left[\hat{\nabla}f_{\mathcal{I}_k}(\mathbf{x}_k^s) - \hat{\nabla}f_{\mathcal{I}_k}(\mathbf{x}_0^s) + \hat{\nabla}f(\mathbf{x}_0^s)\right]$$

$$= \nabla f_\mu(\mathbf{x}_k^s) + \nabla f_\mu(\mathbf{x}_0^s) - \nabla f_\mu(\mathbf{x}_0^s) = \nabla f_\mu(\mathbf{x}_k^s). \tag{51}$$

Combining (50) and (51), we have

$$\mathbb{E}\left[f_\mu(\mathbf{x}_{k+1}^s)\right] \leq \mathbb{E}\left[f_\mu(\mathbf{x}_k^s)\right] - \eta_k\mathbb{E}\left[\|\nabla f_\mu(\mathbf{x}_k^s)\|_2^2\right] + \frac{L}{2}\eta_k^2\mathbb{E}\left[\|\hat{\mathbf{v}}_k^s\|_2^2\right], \tag{52}$$

where the expectation is taken with respect to all random variables.

At RHS of (52), the upper bound on $\mathbb{E}\left[\|\hat{\mathbf{v}}_k^s\|_2^2\right]$ is given by Proposition 1,

$$\mathbb{E}[\|\hat{\mathbf{v}}_k^s\|_2^2] \leq \frac{4(b+18\delta_n)d}{b}\mathbb{E}\left[\|\nabla f(\mathbf{x}_k^s)\|_2^2\right] + \frac{6(4d+1)L^2\delta_n}{b}\mathbb{E}\left[\|\mathbf{x}_k^s - \mathbf{x}_0^s\|_2^2\right]$$

$$+ \frac{(6\delta_n + b)L^2d^2\mu^2}{b} + \frac{72d\sigma^2\delta_n}{b}. \tag{53}$$

In (53), we further bound $\mathbb{E}\left[\|\mathbf{x}_{k+1}^s - \mathbf{x}_0^s\|_2^2\right]$ as

$$\mathbb{E}\left[\|\mathbf{x}_{k+1}^s - \mathbf{x}_0^s\|_2^2\right] = \mathbb{E}\left[\|\mathbf{x}_{k+1}^s - \mathbf{x}_k^s + \mathbf{x}_k^s - \mathbf{x}_0^s\|_2^2\right]$$

$$= \eta_k^2\mathbb{E}\left[\|\hat{\mathbf{v}}_k^s\|_2^2\right] + \mathbb{E}\left[\|\mathbf{x}_k^s - \mathbf{x}_0^s\|_2^2\right] - 2\eta_k\mathbb{E}\left[\langle\hat{\mathbf{v}}_k^s, \mathbf{x}_k^s - \mathbf{x}_0^s\rangle\right]$$

$$\overset{(51)}{=} \eta_k^2\mathbb{E}\left[\|\hat{\mathbf{v}}_k^s\|_2^2\right] + \mathbb{E}\left[\|\mathbf{x}_k^s - \mathbf{x}_0^s\|_2^2\right] - 2\eta_k\mathbb{E}\left[\langle\nabla f_\mu(\mathbf{x}_k^s), \mathbf{x}_k^s - \mathbf{x}_0^s\rangle\right]$$

$$\leq \eta_k^2\mathbb{E}\left[\|\hat{\mathbf{v}}_k^s\|_2^2\right] + \mathbb{E}\left[\|\mathbf{x}_k^s - \mathbf{x}_0^s\|_2^2\right] + 2\eta_k\mathbb{E}\left[\frac{1}{2\beta_k}\|\nabla f_\mu(\mathbf{x}_k^s)\|_2^2 + \frac{\beta_k}{2}\|\mathbf{x}_k^s - \mathbf{x}_0^s\|_2^2\right], \tag{54}$$

where $\beta_k$ is a positive coefficient, and the last inequality holds since $\langle\mathbf{a}, \mathbf{b}\rangle \leq \frac{\beta\|\mathbf{a}\|_2^2 + (1/\beta)\|\mathbf{b}\|_2^2}{2}$ for any $\mathbf{a}$ and $\mathbf{b}$, and $\beta > 0$.

Now with (53) and (54) at hand, we introduce a Lyapunov function [20] with respect to $f_\mu$,

$$R_k^s = \mathbb{E}\left[f_\mu(\mathbf{x}_k^s) + c_k\|\mathbf{x}_k^s - \mathbf{x}_0^s\|_2^2\right], \tag{55}$$

for some $c_k > 0$. Substituting (52) and (54) into $R_{k+1}^s$, we obtain

$$
\begin{aligned}
R_{k+1}^s =& \mathbb{E}\left[f_\mu(\mathbf{x}_{k+1}^s) + c_{k+1}\|\mathbf{x}_{k+1}^s - \mathbf{x}_0^s\|_2^2\right] \\
\leq& \mathbb{E}\left[f_\mu(\mathbf{x}_k^s) - \eta_k\|\nabla f_\mu(\mathbf{x}_k^s)\|_2^2 + \frac{L}{2}\eta_k^2\|\hat{\mathbf{v}}_k^s\|_2^2\right] \\
& + \mathbb{E}\left[c_{k+1}\eta_k^2\|\hat{\mathbf{v}}_k^s\|_2^2 + c_{k+1}\|\mathbf{x}_k^s - \mathbf{x}_0^s\|_2^s\right] \\
& + \mathbb{E}\left[\frac{c_{k+1}\eta_k}{\beta_k}\|\nabla f_\mu(\mathbf{x}_k^s)\|_2^2 + c_{k+1}\beta_k\eta_k\|\mathbf{x}_k^s - \mathbf{x}_0^s\|_2^2\right] \\
=& \mathbb{E}\left[f_\mu(\mathbf{x}_k^s)\right] - \left(\eta_k - \frac{c_{k+1}\eta_k}{\beta_k}\right)\mathbb{E}\left[\|\nabla f_\mu(\mathbf{x}_k^s)\|_2^2\right] \\
& + (c_{k+1} + c_{k+1}\beta_k\eta_k)\mathbb{E}\left[\|\mathbf{x}_k^s - \mathbf{x}_0^s\|_2^2\right] + \left(\frac{L}{2}\eta_k^2 + c_{k+1}\eta_k^2\right)\mathbb{E}\left[\|\hat{\mathbf{v}}_k^s\|_2^2\right]. \tag{56}
\end{aligned}
$$

Moreover, substituting (53) into (56), we have

$$
\begin{aligned}
R_{k+1}^s \leq& \mathbb{E}\left[f_\mu(\mathbf{x}_k^s)\right] - \left(\eta_k - \frac{c_{k+1}\eta_k}{\beta_k}\right)\mathbb{E}\left[\|\nabla f_\mu(\mathbf{x}_k^s)\|_2^2\right] + (c_{k+1} + c_{k+1}\beta_k\eta_k)\mathbb{E}\left[\|\mathbf{x}_k^s - \mathbf{x}_0^s\|_2^2\right] \\
& + \left(\frac{L}{2}\eta_k^2 + c_{k+1}\eta_k^2\right)\frac{6(4d+1)L^2\delta_n}{b}\mathbb{E}\left[\|\mathbf{x}_k^s - \mathbf{x}_0^s\|_2^2\right] \\
& + \left(\frac{L}{2}\eta_k^2 + c_{k+1}\eta_k^2\right)\frac{4db + 72d\delta_n}{b}\mathbb{E}\left[\|\nabla f(\mathbf{x}_k^s)\|_2^2\right] \\
& + \left(\frac{L}{2}\eta_k^2 + c_{k+1}\eta_k^2\right)\frac{(6\delta_n + b)L^2d^2\mu^2 + 72d\sigma^2\delta_n}{b}. \tag{57}
\end{aligned}
$$

Based on the definition of $c_k = c_{k+1} + \beta_k\eta_k c_{k+1} + \frac{6(4d+1)L^2\delta_n\eta_k^2}{b}c_{k+1} + \frac{3(4d+1)L^3\delta_n\eta_k^2}{b}$ and the definition of $R_k^s$ in (55), we can simplify the inequality (57) as

$$
\begin{aligned}
R_{k+1}^s \leq& R_k^s - \left(\eta_k - \frac{c_{k+1}\eta_k}{\beta_k}\right)\mathbb{E}\left[\|\nabla f_\mu(\mathbf{x}_k^s)\|_2^2\right] \\
& + \left(\frac{L}{2}\eta_k^2 + c_{k+1}\eta_k^2\right)\frac{4db + 72d\delta_n}{b}\mathbb{E}\left[\|\nabla f(\mathbf{x}_k^s)\|_2^2\right] \\
& + \left(\frac{L}{2}\eta_k^2 + c_{k+1}\eta_k^2\right)\frac{(6\delta_n + b)L^2d^2\mu^2 + 72d\sigma^2\delta_n}{b} \\
\overset{(18)}{\leq}& R_k^s - \frac{1}{2}\left(\eta_k - \frac{c_{k+1}\eta_k}{\beta_k}\right)\mathbb{E}\left[\|\nabla f(\mathbf{x}_k^s)\|_2^2\right] + \left(\eta_k - \frac{c_{k+1}\eta_k}{\beta_k}\right)\frac{\mu^2d^2L^2}{4} \\
& + \left(\frac{L}{2}\eta_k^2 + c_{k+1}\eta_k^2\right)\frac{4db + 72d\delta_n}{b}\mathbb{E}\left[\|\nabla f(\mathbf{x}_k^s)\|_2^2\right] \\
& + \left(\frac{L}{2}\eta_k^2 + c_{k+1}\eta_k^2\right)\frac{(6\delta_n + b)L^2d^2\mu^2 + 72d\sigma^2\delta_n}{b} \\
=& R_k^s - \gamma_k\mathbb{E}\left[\|\nabla f(\mathbf{x}_k^s)\|_2^2\right] + \chi_k, \tag{58}
\end{aligned}
$$

where $\gamma_k$ and $\chi_k$ are coefficients given by

$$\gamma_k = \frac{1}{2}\left(1 - \frac{c_{k+1}}{\beta_k}\right)\eta_k - \left(\frac{L}{2} + c_{k+1}\right)\frac{4db + 72d\delta_n}{b}\eta_k^2, \tag{59}$$

$$\chi_k = \left(\frac{L}{2} + c_{k+1}\right)\frac{(6\delta_n + b)L^2d^2\mu^2 + 72d\sigma^2\delta_n}{b}\eta_k^2 + \left(1 - \frac{c_{k+1}}{\beta_k}\right)\frac{\mu^2d^2L^2}{4}\eta_k. \tag{60}$$

In the second inequality of (58), we have used the fact that $1 - \frac{c_{k+1}}{\beta_k} > 0$. This holds for some parameter $\eta_k$ under the condition that $\gamma_k > 0$. Even if $1 - \frac{c_{k+1}}{\beta_k} < 0$ (relaxing the condition $\gamma_k > 0$), a

similar inequality can be obtained using the upper bound of $\|\nabla f_\mu(\mathbf{x}_k^s)\|_2^2$ in (18). Therefore, without loss of generality, we consider $1 - \frac{c_{k+1}}{\beta_k} > 0$.

Taking a telescopic sum for (58), we obtain

$$R_m^s \leq R_0^s - \sum_{k=0}^{m-1} \gamma_k \mathbb{E}\left[\|\nabla f(\mathbf{x}_k^s)\|_2^2\right] + \chi_m, \tag{61}$$

where $\chi_m = \sum_{k=0}^{m-1} \chi_k$. It is known from (55) that

$$R_0^s = \mathbb{E}\left[f_\mu(\mathbf{x}_0^s)\right], \quad R_m^s = \mathbb{E}\left[f_\mu(\mathbf{x}_m^s)\right], \tag{62}$$

where the last equality used the fact that $c_m = 0$. Since $\tilde{\mathbf{x}}_{s-1} = \mathbf{x}_0^s$ and $\tilde{\mathbf{x}}_s = \mathbf{x}_m^s$, we obtain

$$R_0^s - R_m^s = \mathbb{E}\left[f_\mu(\tilde{\mathbf{x}}_{s-1}) - f_\mu(\tilde{\mathbf{x}}_s)\right]. \tag{63}$$

Substituting (63) into (61) and telescoping the sum for $s = 1, 2, \ldots, S$, we obtain

$$\sum_{s=1}^{S} \sum_{k=0}^{m-1} \gamma_k \mathbb{E}[\|\nabla f(\mathbf{x}_k^s)\|_2^2] \leq \mathbb{E}[f_\mu(\tilde{\mathbf{x}}_0) - f_\mu(\tilde{\mathbf{x}}_S)] + S\chi_m. \tag{64}$$

Denoting $f_\mu^* = \min_{\mathbf{x}} f_\mu(\mathbf{x})$, from (15) we have $f_\mu(\tilde{\mathbf{x}}_0) - f(\tilde{\mathbf{x}}_0) \leq \frac{\mu^2 L}{2}$ and $f^* - f_\mu^* \leq \frac{\mu^2 L}{2}$, where $f^* = \min_{\mathbf{x}} f(\mathbf{x})$. This yields

$$f_\mu(\tilde{\mathbf{x}}_0) - f_\mu(\tilde{\mathbf{x}}_S) \leq f_\mu(\tilde{\mathbf{x}}_0) - f_\mu^* \leq (f(\tilde{\mathbf{x}}_0) - f^*) + \mu^2 L. \tag{65}$$

Substituting (65) into (64), we have

$$\sum_{s=1}^{S} \sum_{k=0}^{m-1} \gamma_k \mathbb{E}[\|\nabla f(\mathbf{x}_k^s)\|_2^2] \leq \mathbb{E}[f(\tilde{\mathbf{x}}_0) - f^*] + L\mu^2 + S\chi_m. \tag{66}$$

Let $\bar{\gamma} = \min_k \gamma_k$ and we choose $\bar{\mathbf{x}}$ uniformly random from $\{\{\mathbf{x}_k^s\}_{k=0}^{m-1}\}_{s=1}^{S}$, then ZO-SVRG satisfies

$$\mathbb{E}[\|\nabla f(\bar{\mathbf{x}})\|_2^2] \leq \frac{\mathbb{E}[f(\tilde{\mathbf{x}}_0) - f^*]}{T\bar{\gamma}} + \frac{L\mu^2}{T\bar{\gamma}} + \frac{S\chi_m}{T\bar{\gamma}}. \tag{67}$$

The proof is now complete. $\qquad\qquad\square$

## A.5   Proof of Corollary 1

We start by rewriting $c_k$ in (8) as

$$c_k = (1+\theta)c_{k+1} + \frac{3(1+4d)L^3\delta_n\eta^2}{b}, \tag{68}$$

where $\theta = \beta\eta + \frac{6(1+4d)L^2\delta_n\eta^2}{b}$. The recursive formula (68) implies that $c_k \leq c_0$ for any $k$, and

$$c_0 = \frac{3(1+4d)L^3\delta_n\eta^2}{b} \frac{(1+\theta)^m - 1}{\theta}. \tag{69}$$

Based on the choice of $\eta = \frac{\rho}{Ld}$ and $\beta = L$, we have

$$\theta = \frac{\rho}{d} + \frac{6\rho^2\delta_n}{bd^2} + \frac{24\rho^2\delta_n}{bd} \leq \frac{31\rho}{d}, \tag{70}$$

where we have used the fact that $\delta_n \leq 1$. Substituting (70) into (69), we have

$$\begin{aligned}
c_k \leq c_0 &= \frac{3(1+4d)L^3\delta_n}{b}\frac{\eta^2}{\theta}[(1+\theta)^m - 1] = \frac{3(1+4d)L\rho\delta_n}{db + 24\rho d + 6\rho}[(1+\theta)^m - 1] \\
&\leq \frac{15dL\rho\delta_n}{db}[(1+\theta)^m - 1] \leq \frac{15L\rho\delta_n}{b}(e-1) \leq \frac{30L\rho\delta_n}{b},
\end{aligned} \tag{71}$$

where the third inequality holds since $(1 + \theta)^m \leq (1 + \frac{31\rho}{d})^m$, $m = \lceil \frac{d}{31\rho} \rceil$, $(1 + 1/a)^a \leq \lim_{a\to\infty}(1 + \frac{1}{a})^a = e$ for $a > 0$ [20, Appendix E], and for east of representation, the last inequality loosely uses the notion '$\leq$' since $e < 3$.

We recall from (5) and (6) that

$$\bar{\gamma} = \min_{0 \leq k \leq m-1} \left\{ \frac{\eta_k}{2} - \frac{c_{k+1}\eta_k}{2\beta_k} - \eta_k^2 \left( \frac{L}{2} + c_{k+1} \right) \left( 4d + \frac{72d\delta_n}{b} \right) \right\}. \tag{72}$$

Since $\eta_k = \eta$, $\beta_k = \beta$, and $c_k \leq c_0$, we have

$$\bar{\gamma} \geq \frac{\eta}{2} - \frac{c_0}{2\beta}\eta - \eta^2 L \left( 2d + \frac{36d}{b} \right) - \eta^2 c_0 \left( 4d + \frac{72d\delta_n}{b} \right). \tag{73}$$

From (71) and the definition of $\beta$, we have

$$\frac{c_0}{2\beta} \leq \frac{15\rho}{b} \tag{74}$$

$$\eta L \left( 2d + \frac{36d}{b} \right) = \rho \left( 2 + \frac{36}{b} \right) \tag{75}$$

$$\eta c_0 \left( 4d + \frac{72d\delta_n}{b} \right) \overset{(71)}{\leq} \frac{\rho}{Ld} \frac{30L\rho}{b} \left( 4d + \frac{72d}{b} \right) \leq \frac{120\rho^2}{b} + \frac{2160\rho^2}{b^2}. \tag{76}$$

Substituting (74)-(76) into (73), we obtain

$$\bar{\gamma} \geq \eta \left( \frac{1}{2} - \frac{15\rho}{b} - 4\rho - \frac{240\rho^2}{b} \right) \geq \eta \left( \frac{1}{2} - 259\rho \right), \tag{77}$$

where we have used the fact that $\rho^2 \leq \rho$. Moreover, if we set $\rho \leq \frac{1}{518}$, then $\bar{\gamma} > 0$. In other words, the current parameter setting is valid for Theorem 1. Upon defining a universal constant $\alpha_0 = \left( \frac{1}{2} - 259\rho \right)$, we have

$$\bar{\gamma} \geq \eta \alpha_0. \tag{78}$$

Next, we find the upper bound on $\chi_m$ in (7) given the current parameter setting and $c_k \leq c_0$,

$$\chi_m \leq m\eta \frac{\mu^2 d^2 L^2}{4} + m\eta^2 \left( \frac{L}{2} + c_0 \right) \frac{72d\sigma^2\delta_n + (6\delta_n + b)L^2 d^2 \mu^2}{b}. \tag{79}$$

Since $\frac{L}{2} + c_0 \leq \frac{L}{2} + 30L\rho b^{-1} \leq \frac{L}{2} + 2L = \frac{5L}{2}$ (suppose $b \geq 18$ without loss of generality), based on (78) we have

$$\frac{\chi_m}{\bar{\gamma}} \leq m \frac{d^2 L^2 \mu^2}{4\alpha_0} + m \frac{5L}{2\alpha_0} \frac{72d\sigma^2\delta_n}{b} \frac{\rho}{Ld} + m \frac{5L}{2\alpha_0} \left( \frac{6L^2 d^2 \mu^2 \delta_n}{b} + L^2 d^2 \mu^2 \right) \frac{\rho}{Ld}. \tag{80}$$

Since $T = Sm$, and $\mu = \frac{1}{\sqrt{dT}}$, the above inequality yields

$$\frac{S\chi_m}{T\bar{\gamma}} \leq \frac{dL^2}{4\alpha_0 T} + \frac{180\sigma^2 \rho \delta_n}{b\alpha_0} + \frac{5L^2}{2\alpha_0} \left( \frac{6}{b} + 1 \right) \frac{\rho}{T} = O \left( \frac{d}{T} + \frac{\delta_n}{b} \right), \tag{81}$$

where in the big $O$ notation, we only keep the dominant terms and ignore the constant numbers that are independent of $d$, $b$, and $T$.

Substituting (78) and (81) into (5), we have

$$\mathbb{E}[\|\nabla f(\bar{\mathbf{x}})\|_2^2] \leq \frac{[f(\tilde{\mathbf{x}}_0) - f^*]}{T\alpha_0} \frac{Ld}{\rho} + \frac{L^2}{T^2 \alpha_0 \rho} + \frac{S\chi_m}{T\bar{\gamma}} = O \left( \frac{d}{T} + \frac{\delta_n}{b} \right). \tag{82}$$

The proof is now complete. $\qquad\qquad\qquad\qquad\qquad\qquad\qquad\qquad\qquad\qquad\qquad\qquad\qquad\square$

## A.6 Proof of Proposition 2

For RandGradEst, based on (17) and (19), we have

$$\mathbb{E}\left[\|\hat{\nabla}f(\mathbf{x}) - \nabla f(\mathbf{x})\|_2^2\right] \leq \mathbb{E}\left[\|\hat{\nabla}f(\mathbf{x}) - \nabla f_\mu(\mathbf{x}) + \nabla f_\mu(\mathbf{x}) - \nabla f(\mathbf{x})\|_2^2\right]$$

$$\leq 2\mathbb{E}\left[\|\hat{\nabla}f(\mathbf{x}) - \nabla f_\mu(\mathbf{x})\|_2^2\right] + 2\|\nabla f_\mu(\mathbf{x}) - \nabla f(\mathbf{x})\|_2^2$$

$$\leq 4d\|\nabla f(\mathbf{x})\|_2^2 + \frac{3\mu^2 L^2 d^2}{2} = O\left(d\|\nabla f(\mathbf{x})\|_2^2 + \mu^2 L^2 d^2\right). \tag{83}$$

Similarly, for Avg-RandGradEst, based on (17) and (25), we have

$$\mathbb{E}\left[\|\hat{\nabla}f(\mathbf{x}) - \nabla f(\mathbf{x})\|_2^2\right] \leq 4\left(1 + \frac{d}{q}\right)\|\nabla f(\mathbf{x})\|_2^2 + \left(3 + \frac{2}{q}\right)\frac{\mu^2 L^2 d^2}{2}$$

$$= O\left(\frac{q+d}{q}\|\nabla f(\mathbf{x})\|_2^2 + \mu^2 L^2 d^2\right), \tag{84}$$

where we have used the fact that $\frac{2}{q} \leq 3$.

Finally, using (31), the proof is then complete. $\qquad\square$

## A.7 Proof of Theorem 2

Motivated by Proposition 1, we first bound $\|\hat{\mathbf{v}}_k^s\|_2^2$. Following (36)-(42), we have

$$\mathbb{E}\left[\|\hat{\mathbf{v}}_k^s\|_2^2\right] \leq \frac{2\delta_n}{bn}\sum_{i=1}^n \mathbb{E}\left[\|\hat{\nabla}f_i(\mathbf{x}_k^s) - \hat{\nabla}f_i(\mathbf{x}_0^s)\|_2^2\right] + 2\mathbb{E}\left[\|\hat{\nabla}f(\mathbf{x}_k^s)\|_2^2\right]$$

$$\overset{(25)}{\leq} \frac{2\delta_n}{bn}\sum_{i=1}^n \mathbb{E}\left[\|\hat{\nabla}f_i(\mathbf{x}_k^s) - \hat{\nabla}f_i(\mathbf{x}_0^s)\|_2^2\right] + 4\left(1 + \frac{d}{q}\right)\|\nabla f(\mathbf{x}_k^s)\|_2^2 + \left(1 + \frac{1}{q}\right)\mu^2 L^2 d^2. \tag{85}$$

Moreover, following (43)-(47) together with (25), we can obtain that

$$\mathbb{E}\left[\|\hat{\nabla}f_i(\mathbf{x}_k^s) - \hat{\nabla}f_i(\mathbf{x}_0^s)\|_2^2\right]$$

$$\leq 36\left(1 + \frac{d}{q}\right)\sigma^2 + 36\left(1 + \frac{d}{q}\right)\mathbb{E}\left[\|\nabla f(\mathbf{x}_k^s)\|_2^2\right] + \left(12\frac{d}{q} + 15\right)L^2\|\mathbf{x}_k^s - \mathbf{x}_0^s\|_2^2$$

$$+ 3\left(1 + \frac{1}{q}\right)L^2\mu^2 d^2. \tag{86}$$

Substituting (86) into (85), we have

$$\mathbb{E}\left[\|\hat{\mathbf{v}}_k^s\|_2^2\right] \leq \frac{4(b + 18\delta_n)}{b}\left(1 + \frac{d}{q}\right)\mathbb{E}[\|\nabla f(\mathbf{x}_k^s)\|_2^2] + \frac{6\delta_n}{b}\left(\frac{4d}{q} + 5\right)L^2\|\mathbf{x}_k^s - \mathbf{x}_0^s\|_2^2$$

$$+ \frac{6\delta_n + b}{b}\left(1 + \frac{1}{q}\right)L^2\mu^2 d^2 + \frac{72\delta_n}{b}\left(1 + \frac{d}{q}\right)\sigma^2. \tag{87}$$

Following (54)-(56) and substituting (87) into (56), we have

$$R_{k+1}^s \leq \mathbb{E}\left[f_\mu(\mathbf{x}_k^s)\right] - \left(\eta_k - \frac{c_{k+1}\eta_k}{\beta_k}\right)\mathbb{E}\left[\|\nabla f_\mu(\mathbf{x}_k^s)\|_2^2\right] + (c_{k+1} + c_{k+1}\beta_k\eta_k)\mathbb{E}\left[\|\mathbf{x}_k^s - \mathbf{x}_0^s\|_2^2\right]$$

$$+ \left(\frac{L}{2}\eta_k^2 + c_{k+1}\eta_k^2\right)\frac{6(4d + 5q)L^2\delta_n}{bq}\mathbb{E}\left[\|\mathbf{x}_k^s - \mathbf{x}_0^s\|_2^2\right]$$

$$+ \left(\frac{L}{2}\eta_k^2 + c_{k+1}\eta_k^2\right)\frac{(72\delta_n + 4b)(q + d)}{bq}\mathbb{E}\left[\|\nabla f(\mathbf{x}_k^s)\|_2^2\right]$$

$$+ \left(\frac{L}{2}\eta_k^2 + c_{k+1}\eta_k^2\right)\frac{(6\delta_n + b)(q + 1)L^2 d^2\mu^2 + 72(q + d)\sigma^2\delta_n}{bq}. \tag{88}$$

Based on the definitions of $c_k = \left[1 + \beta_k \eta_k + \frac{6(4d+5q)L^2 \delta_n}{bq} \eta_k^2\right] c_{k+1} + \frac{3(4d+5q)L^3 \delta_n}{bq} \eta_k^2$ and $R_k^s$ given by (55), we can simplify (88) to

$$R_{k+1}^s \overset{(18)}{\leq} R_k^s - \frac{1}{2}\left(\eta_k - \frac{c_{k+1}\eta_k}{\beta_k}\right) \mathbb{E}\left[\|\nabla f(\mathbf{x}_k^s)\|_2^2\right] + \left(\eta_k - \frac{c_{k+1}\eta_k}{\beta_k}\right)\frac{\mu^2 d^2 L^2}{4}$$
$$+ \left(\frac{L}{2}\eta_k^2 + c_{k+1}\eta_k^2\right)\frac{(72\delta_n + 4b)(q+d)}{bq}\mathbb{E}\left[\|\nabla f(\mathbf{x}_k^s)\|_2^2\right]$$
$$+ \left(\frac{L}{2}\eta_k^2 + c_{k+1}\eta_k^2\right)\frac{(6\delta_n + b)(q+1)L^2 d^2 \mu^2 + 72(q+d)\sigma^2 \delta_n}{bq}$$
$$= R_k^s - \gamma_k \mathbb{E}\left[\|\nabla f(\mathbf{x}_k^s)\|_2^2\right] + \chi_k, \tag{89}$$

where $\gamma_k$ and $\chi_k$ are defined coefficients in Theorem 2.

Based on (89) and following the same argument in (61)-(67), we then achieve

$$\mathbb{E}[\|\nabla f(\bar{\mathbf{x}})\|_2^2] \leq \frac{\mathbb{E}[f(\tilde{\mathbf{x}}_0) - f^*]}{T\bar{\gamma}} + \frac{L\mu^2}{T\bar{\gamma}} + \frac{S\chi_m}{T\bar{\gamma}}. \tag{90}$$

The rest of the proofs essentially follow along the lines of Corollary 1 with the added complexity of the mini-batch parameter $q$ in $c_k$, $\gamma_k$ and $\chi_k$.

Let $\theta = \beta_k \eta_k + \frac{6(4d+5q)L^2 \delta_n}{bq}\eta_k^2$, then $c_k = c_{k+1}(1+\theta) + \frac{3(4d+5q)L^3 \eta_k^2 \delta_n}{bq}$. This leads to

$$c_0 = \frac{3(4d+5q)L^3 \eta^2 \delta_n}{bq}\frac{(1+\theta)^m - 1}{\theta}. \tag{91}$$

Based on the choice of $\eta$ and $\beta$, we have

$$\theta = \frac{\rho}{d} + \frac{24\rho^2 \delta_n}{bdq} + \frac{30\rho^2 \delta_n}{bd^2} \leq \frac{55\rho}{d}. \tag{92}$$

Substituting (92) into (91), we have

$$c_k \leq c_0 = \delta_n \frac{3(5+4d/q)L^3}{b}\frac{\eta^2}{\theta}[(1+\theta)^m - 1] = \delta_n \frac{3(5+4d/q)L\rho}{db + 24\rho d/q + 30\rho}[(1+\theta)^m - 1]$$
$$\leq \delta_n \frac{3(5+4d/q)L\rho}{db}[(1+\theta)^m - 1] \leq \delta_n \frac{27L\rho}{b\min\{d,q\}}[(1+\theta)^m - 1]$$
$$\leq \frac{27L\rho\delta_n}{b\min\{d,q\}}(e-1) \leq \frac{54L\rho\delta_n}{b\min\{d,q\}}, \tag{93}$$

where the third inequality holds since $5 + 4d/q \leq 9d/q$ if $d \geq q$, and $5 + 4d/q \leq 9$ otherwise, and the forth inequality holds similar to (71) under $m = \lceil \frac{d}{55\rho}\rceil$.

According to the definition of $\bar{\gamma} = \min_k \gamma_k$, we have

$$\bar{\gamma} \geq \frac{\eta}{2} - \frac{c_0}{2\beta}\eta - \eta^2 L\frac{(36\delta_n + 2b)(q+d)}{bq} - \eta^2 c_0 \frac{(72\delta_n + 4b)(q+d)}{bq}. \tag{94}$$

From (93) and the definition of $\beta = L$, we have

$$\frac{c_0}{2\beta} \leq \frac{27\rho}{b\min\{d,q\}}. \tag{95}$$

Since $\eta = \rho/(Ld)$, we have

$$\eta L\frac{(36\delta_n + 2b)(q+d)}{bq} \leq \frac{2\rho}{\min\{d,q\}}\left(\frac{36}{b} + 2\right), \tag{96}$$

where we used the fact that $\frac{1}{d} + \frac{1}{q} \leq \frac{2}{\min\{d,q\}}$. Moreover, we have

$$\eta c_0 \frac{(72\delta_n + 4b)(q+d)}{bq} \leq \frac{\rho}{L}\frac{54L\rho}{b\min\{q,d\}}\left(4 + \frac{72}{b}\right)\left(\frac{1}{d} + \frac{1}{q}\right)$$
$$\leq \frac{108\rho^2}{b\min\{d,q\}^2}\left(4 + \frac{72}{b}\right) \tag{97}$$

Substituting (95)-(97) into (94), and following the arguments in (78), we obtain

$$\bar{\gamma} \geq \alpha_0 \eta, \tag{98}$$

where $\alpha_0 > 0$ is a universal constant that is independent of $T$, $d$ and $b$.

Based on $\chi_k = \left(1 - \frac{c_{k+1}}{\beta_k}\right) \frac{\mu^2 d^2 L^2}{4} \eta_k + \left(\frac{L}{2} + c_{k+1}\right) \frac{(6\delta_n + b)(q+1)L^2 d^2 \mu^2 + 72(q+d)\sigma^2 \delta_n}{bq} \eta_k^2$, the upper bound on $\chi_m = \sum_k \chi_k$ is given by

$$\chi_m \leq \eta m \frac{\mu^2 d^2 L^2}{4} + \eta m \left(\frac{L}{2} + c_0\right) \frac{(6\delta_n + b)(q+1)L^2 d^2 \mu^2 + 72(q+d)\sigma^2 \delta_n}{bq} \eta. \tag{99}$$

Using (93) and assuming $b \geq 18$ (without loss of generality), then $\frac{L}{2} + c_0 \leq \frac{L}{2} + 54L\rho b^{-1} \leq \frac{7L}{2}$. This yields

$$\frac{\chi_m}{\bar{\gamma}} \leq \frac{m}{\alpha_0} \frac{d^2 L^2}{4} \frac{1}{dT} + \frac{m}{\alpha_0} \frac{7L}{2} \frac{(6\delta_n + b)(q+1)L^2 d^2}{bq} \frac{1}{dT} \frac{\rho}{Ld} + \frac{m}{\alpha_0} \frac{7L}{2} \frac{72\sigma^2}{b} \left(\frac{1}{d} + \frac{1}{q}\right) \rho \delta_n$$

$$\leq O\left(\frac{md}{T} + \frac{m\delta_n}{b \min\{d, q\}}\right) \tag{100}$$

Since $T = Sm$, we have

$$\frac{S\chi_m}{T\bar{\gamma}} \leq O\left(\frac{d}{T} + \frac{\delta_n}{b \min\{d, q\}}\right). \tag{101}$$

Substituting (98) and (101) into (5), we have

$$\mathbb{E}[\|\nabla f(\bar{\mathbf{x}})\|_2^2] \leq \frac{[f(\tilde{\mathbf{x}}_0) - f^*]}{T\alpha_0} \frac{Ld}{\rho} + \frac{L^2}{T^2 \alpha_0 \rho} + \frac{S\chi_m}{T\bar{\gamma}} = O\left(\frac{d}{T} + \frac{\delta_n}{b \min\{d, q\}}\right). \tag{102}$$

$$\square$$

## A.8 Proof of Theorem 3

Since $f$ is $L$-smooth, we have

$$f(\mathbf{x}_{k+1}^s) \leq f(\mathbf{x}_k^s) - \eta_k \langle \nabla f(\mathbf{x}_k^s), \hat{\mathbf{v}}_k^s \rangle + \frac{L}{2} \eta_k^2 \|\hat{\mathbf{v}}_k^s\|_2^2. \tag{103}$$

Since $\mathbf{x}_k^s$ and $\mathbf{x}_0^s$ are independent of $\mathcal{I}_k$ used in $\hat{\nabla} f_{\mathcal{I}_k}(\mathbf{x}_k^s)$ and $\hat{\nabla} f_{\mathcal{I}_k}(\mathbf{x}_0^s)$, we obtain

$$\mathbb{E}_{\mathcal{I}_k}[\mathbf{v}_k^s] = \hat{\nabla} f(\mathbf{x}_k^s) + \hat{\nabla} f(\mathbf{x}_0^s) - \hat{\nabla} f(\mathbf{x}_0^s) = \hat{\nabla} f(\mathbf{x}_k^s), \tag{104}$$

where we recall that a deterministic gradient estimator is used. Combining (103) and (104), we have

$$\mathbb{E}\left[f(\mathbf{x}_{k+1}^s)\right] \leq \mathbb{E}\left[f(\mathbf{x}_k^s)\right] - \eta_k \mathbb{E}\left[\langle \nabla f(\mathbf{x}_k^s), \hat{\nabla} f(\mathbf{x}_k^s) \rangle\right] + \frac{L}{2} \eta_k^2 \mathbb{E}\left[\|\hat{\mathbf{v}}_k^s\|_2^2\right]. \tag{105}$$

In (105), we bound $-2\mathbb{E}\left[\langle \nabla f(\mathbf{x}_k^s), \hat{\nabla} f(\mathbf{x}_k^s) \rangle\right]$ as,

$$-2\mathbb{E}\left[\langle \nabla f(\mathbf{x}_k^s), \hat{\nabla} f(\mathbf{x}_k^s) \rangle\right] \leq \mathbb{E}\left[\|\nabla f(\mathbf{x}_k^s) - \hat{\nabla} f(\mathbf{x}_k^s)\|_2^2\right] - \left[\mathbb{E}\|\nabla f(\mathbf{x}_k^s)\|_2^2\right]$$

$$\stackrel{(31)}{\leq} \frac{L^2 d^2 \mu^2}{4} - \mathbb{E}\left[\|\nabla f(\mathbf{x}_k)\|_2^2\right], \tag{106}$$

where the first inequality holds since $-2\langle \mathbf{a}, \mathbf{b} \rangle \leq \|\mathbf{a} - \mathbf{b}\|_2^2 - \|\mathbf{a}\|_2^2$, and we have used the fact that $\mu_\ell = \mu$ in the second inequality.

Substituting (106) into (105), we have

$$\mathbb{E}\left[f(\mathbf{x}_{k+1}^s)\right] \leq \mathbb{E}\left[f(\mathbf{x}_k^s)\right] - \frac{\eta_k}{2} \mathbb{E}\left[\|\nabla f(\mathbf{x}_k)\|_2^2\right] + \frac{L}{2} \eta_k^2 \mathbb{E}\left[\|\hat{\mathbf{v}}_k^s\|_2^2\right] + \frac{L^2 d^2 \mu^2 \eta_k}{8}. \tag{107}$$

In (107), we next bound $\mathbb{E}\left[\|\hat{\mathbf{v}}_k^s\|_2^2\right]$. Following (36)-(42), we have

$$\mathbb{E}\left[\|\hat{\mathbf{v}}_k^s\|_2^2\right] \leq \frac{2\delta_n}{bn} \sum_{i=1}^n \mathbb{E}\left[\|\hat{\nabla} f_i(\mathbf{x}_k^s) - \hat{\nabla} f_i(\mathbf{x}_0^s)\|_2^2\right] + 2\mathbb{E}\left[\|\hat{\nabla} f(\mathbf{x}_k^s)\|_2^2\right]. \tag{108}$$

The first term at RHS of (108) yields

$$\mathbb{E}\left[\|\hat{\nabla} f_i(\mathbf{x}_k^s) - \hat{\nabla} f_i(\mathbf{x}_0^s)\|_2^2\right] \overset{(28)}{=} \mathbb{E}\left[\left\|\sum_{\ell=1}^d \left(\frac{\partial f_{i,\mu_\ell}}{\partial x_{k,\ell}^s}\mathbf{e}_\ell - \frac{\partial f_{i,\mu_\ell}}{\partial x_{0,\ell}^s}\mathbf{e}_\ell\right)\right\|_2^2\right]$$

$$\overset{(130)}{\leq} d\sum_{\ell=1}^d \mathbb{E}\left[\left\|\frac{\partial f_{i,\mu_\ell}}{\partial x_{k,\ell}^s} - \frac{\partial f_{i,\mu_\ell}}{\partial x_{0,\ell}^s}\right\|_2^2\right] \leq L^2 d\sum_{\ell=1}^d \mathbb{E}\left[\|x_{\ell,k}^s - x_{\ell,0}^s\|_2^2\right] = L^2 d\mathbb{E}\left[\|\mathbf{x}_k^s - \mathbf{x}_0^s\|_2^2\right], \tag{109}$$

where $f_{i,\mu_\ell}(\mathbf{x}) = \mathbb{E}_{u\sim U[-\mu_\ell,\mu_\ell]} f_i(\mathbf{x} + u\mathbf{e}_\ell)$ denotes the smooth function of $f_i$ with respect to its $\ell$th coordinate (Lemma 3), $x_{k,\ell}^s$ denotes the $\ell$th coordinate of $\mathbf{x}_k^s$, $\frac{\partial f_{i,\mu_\ell}}{\partial x_{k,\ell}^s}$ is the $\ell$th partial derivative of $f_{i,\mu_\ell}$ at $\mathbf{x}_k^s$, and the second inequality holds since $f_{i,\mu_\ell}(\mathbf{x})$ is $L$-smooth (Lemma 3) with respect to the $\ell$th coordinate. From (31), the second term at RHS of (108) yields

$$\|\hat{\nabla} f(\mathbf{x})\|_2^2 \leq 2\|\nabla f(\mathbf{x})\|_2^2 + 2\|\hat{\nabla} f(\mathbf{x}) - \nabla f(\mathbf{x})\|_2^2 \overset{(31)}{\leq} 2\|\nabla f(\mathbf{x})\|_2^2 + \frac{L^2 d^2 \mu^2}{2}, \tag{110}$$

where we have used the fact that $\mu_\ell = \mu$.

Substituting (109) and (110) into (108), we have

$$\mathbb{E}\left[\|\hat{\mathbf{v}}_k^s\|_2^2\right] \leq \frac{2L^2 d\delta_n}{b}\mathbb{E}\left[\|\mathbf{x}_k^s - \mathbf{x}_0^s\|_2^2\right] + 4\mathbb{E}\left[\|\nabla f(\mathbf{x})\|_2^2\right] + L^2 d^2 \mu^2. \tag{111}$$

Similar to (54), we have

$$\mathbb{E}\left[\|\mathbf{x}_{k+1}^s - \mathbf{x}_0^s\|_2^2\right] \leq \eta_k^2 \mathbb{E}\left[\|\hat{\mathbf{v}}_k^s\|_2^2\right] + \mathbb{E}\left[\|\mathbf{x}_k^s - \mathbf{x}_0^s\|_2^2\right] + \eta_k \mathbb{E}\left[\frac{1}{\beta_k}\|\hat{\nabla} f(\mathbf{x}_k^s)\|_2^2 + \beta_k\|\mathbf{x}_k^s - \mathbf{x}_0^s\|_2^2\right]$$

$$\overset{(110)}{\leq} \eta_k^2 \mathbb{E}\left[\|\hat{\mathbf{v}}_k^s\|_2^2\right] + \mathbb{E}\left[\|\mathbf{x}_k^s - \mathbf{x}_0^s\|_2^2\right] + \eta_k \mathbb{E}\left[\frac{2}{\beta_k}\|\nabla f(\mathbf{x}_k^s)\|_2^2 + \beta_k\|\mathbf{x}_k^s - \mathbf{x}_0^s\|_2^2\right] + \frac{L^2 \mu^2 d^2 \eta_k}{\beta_k 2}. \tag{112}$$

Define the following Lyapunov function [20],

$$R_k^s = \mathbb{E}\left[f(\mathbf{x}_k^s) + c_k\|\mathbf{x}_k^s - \mathbf{x}_0^s\|_2^2\right], \tag{113}$$

where $c_k > 0$.

Based on (107) and (112), we obtain

$$R_{k+1}^s = \mathbb{E}\left[f(\mathbf{x}_{k+1}^s) + c_{k+1}\|\mathbf{x}_{k+1}^s - \mathbf{x}_0^s\|_2^2\right]$$

$$\leq \mathbb{E}[f(\mathbf{x}_k^s)] - \left(\frac{\eta_k}{2} - \frac{c_{k+1}\eta_k}{2\beta_k}\right)\mathbb{E}\left[\|\nabla f(\mathbf{x}_k^s)\|_2^2\right] + (c_{k+1} + c_{k+1}\beta_k\eta_k)\mathbb{E}\left[\|\mathbf{x}_k^s - \mathbf{x}_0^s\|_2^2\right]$$

$$+ \left(\frac{L}{2}\eta_k^2 + c_{k+1}\eta_k^2\right)\mathbb{E}\left[\|\hat{\mathbf{v}}_k^s\|_2^2\right] + \frac{d^2 L^2 \mu^2 \eta_k}{8} + \frac{d^2 L^2 \mu^2 c_{k+1}\eta_k}{2\beta_k}. \tag{114}$$

Substituting (111) into (114), we have

$$R_{k+1}^s \leq \mathbb{E}\left[f(\mathbf{x}_k^s)\right] - \left(\frac{\eta_k}{2} - \frac{c_{k+1}\eta_k}{2\beta_k}\right)\mathbb{E}\left[\|\nabla f(\mathbf{x}_k^s)\|_2^2\right] + (c_{k+1} + c_{k+1}\beta_k\eta_k)\mathbb{E}\left[\|\mathbf{x}_k^s - \mathbf{x}_0^s\|_2^2\right]$$

$$+ \left(\frac{L}{2}\eta_k^2 + c_{k+1}\eta_k^2\right)\frac{2L^2 d\delta_n}{b}\mathbb{E}\left[\|\mathbf{x}_k^s - \mathbf{x}_0^s\|_2^2\right] + \left(\frac{L}{2}\eta_k^2 + c_{k+1}\eta_k^2\right)4\mathbb{E}\left[\|\nabla f(\mathbf{x}_k^s)\|_2^2\right]$$

$$+ \left(\frac{L}{2}\eta_k^2 + c_{k+1}\eta_k^2\right)\mu^2 L^2 d^2 + \frac{d^2 L^2 \mu^2 \eta_k}{8} + \frac{d^2 L^2 \mu^2 c_{k+1}\eta_k}{2\beta_k}. \tag{115}$$

Based on the definition of $c_k$, i.e.,

$$c_k = \left(1 + \beta_k \eta_k + \frac{2dL^2 \eta_k^2 \delta_n}{b}\right) c_{k+1} + \frac{dL^3 \eta_k^2 \delta_n}{b},$$

we can simplify (115) to

$$R_{k+1}^s \overset{(113)}{\leq} R_k^s - \gamma_k \mathbb{E}[\|\nabla f(\mathbf{x}_k^s)\|_2^2] + \chi_k, \tag{116}$$

where we recall that

$$\gamma_k = \frac{1}{2}\left(1 - \frac{c_{k+1}}{\beta_k}\right)\eta_k - 4\left(\frac{L}{2} + c_{k+1}\right)\eta_k^2,$$

$$\chi_k = \frac{1}{2}\left(\frac{1}{4} + \frac{c_{k+1}}{\beta_k}\right)L^2 d^2 \mu^2 \eta_k + \left(\frac{L}{2} + c_{k+1}\right)\mu^2 L^2 d^2 \eta_k^2.$$

Based on (116) and following the similar argument in (61)-(64), we have

$$\sum_{s=1}^S \sum_{k=0}^{m-1} \gamma_k \mathbb{E}[\|\nabla f(\mathbf{x}_k^s)\|_2^2] \leq \mathbb{E}[f(\tilde{\mathbf{x}}_0) - f^*] + S\chi_m.$$

Consider $\bar{\gamma} = \min_k \gamma_k$ and the distribution of choosing $\bar{\mathbf{x}}$, we obtain

$$\mathbb{E}[\|\nabla f(\bar{\mathbf{x}})\|_2^2] \leq \frac{\mathbb{E}[f(\tilde{\mathbf{x}}_0) - f^*]}{T\bar{\gamma}} + \frac{S\chi_m}{T\bar{\gamma}}. \tag{117}$$

The rest of the proofs essentially follow along the lines of Corollary 1 under a different parameter setting.

Since $c_k = c_{k+1}(1+\theta) + \frac{dL^3\eta^2\delta_n}{b}$, we have $c_k \leq c_0$ for any $k$, and $\theta = \beta\eta + \frac{2dL^2\eta^2\delta_n}{b}$. This yields

$$c_0 = \frac{dL^3\eta^2\delta_n}{b}\frac{(1+\theta)^m - 1}{\theta}. \tag{118}$$

When $\eta = \rho/(Ld)$ and $\beta = L$ we have

$$\theta = \frac{\rho}{d} + \frac{2\rho^2\delta_n}{bd} \leq \frac{3\rho}{d}. \tag{119}$$

Substituting (119) into (118), we have

$$c_k \leq c_0 = \delta_n \frac{dL^3}{b}\frac{\eta^2}{\theta}[(1+\theta)^m - 1] = \delta_n \frac{\rho L}{b + 2\rho}[(1+\theta)^m - 1] \leq \delta_n \frac{L\rho}{b}(e-1) \leq \delta_n \frac{2L\rho}{b}, \tag{120}$$

where the second equality holds similar to (71) under $m = \lceil\frac{d}{3\rho}\rceil$.

Based on (120) and the definition of $\bar{\gamma}$, similar to (74)-(78) we can obtain

$$\bar{\gamma} \geq \eta\alpha_0, \tag{121}$$

where $\alpha_0 > 0$ is independent of $T$, $d$ and $b$.

Since $\chi_m = \sum_k \chi_k$, it can be bounded as

$$\chi_m \leq m\eta^2 \left(\frac{L}{2} + c_0\right)\mu^2 L^2 d^2 + m\eta\frac{d^2 L^2 \mu^2}{8} + m\eta\frac{d^2 L^2 \mu^2 c_0}{2\beta}. \tag{122}$$

From (120), we have $\frac{L}{2} + c_0 \leq \frac{L}{2} + 2L\rho b^{-1}\delta_n \leq \frac{5L}{2}$. Moreover, based on $T = Sm$ and $\mu = \frac{1}{\sqrt{d}\sqrt{T}}$, we have

$$\frac{S\chi_m}{T\bar{\gamma}} \leq \frac{5L^2\rho}{2\alpha_0 T} + \frac{dL^2}{8\alpha_0 T} + \frac{d\rho L^2}{\alpha_0 bT} = O\left(\frac{1}{T} + \frac{d}{T} + \frac{d}{bT}\right), \tag{123}$$

where in the big $O$ notation, we ignore the constant numbers that are independent of $L$, $d$, $b$, and $T$.

Substituting (121) and (123) into (13), we have

$$\mathbb{E}[\|\nabla f(\bar{\mathbf{x}})\|_2^2] \leq \frac{[f(\tilde{\mathbf{x}}_0) - f^*]}{T\alpha_0}\frac{Ld}{\rho} + \frac{S\chi_m}{T\bar{\gamma}} = O\left(\frac{d}{T}\right). \tag{124}$$

$\square$

## A.9 Auxiliary Lemmas

**Lemma 4** *Let $\{\mathbf{z}_i\}_{i=1}^n$ be a sequence of $n$ vectors. Let $\mathcal{I}$ be a mini-batch of size $b$, which contains i.i.d. samples selected uniformly randomly (with replacement) from $[n]$. Then*

$$\mathbb{E}_{\mathcal{I}}\left[\frac{1}{b}\sum_{i\in\mathcal{I}}\mathbf{z}_i\right] = \frac{1}{n}\sum_{j=1}^n\mathbf{z}_j. \tag{125}$$

*When $\sum_{i=1}^n\mathbf{z}_i = \mathbf{0}$, then*

$$\mathbb{E}_{\mathcal{I}}\left[\left\|\frac{1}{b}\sum_{i\in\mathcal{I}}\mathbf{z}_i\right\|_2^2\right] = \frac{1}{bn}\sum_{i=1}^n\|\mathbf{z}_i\|_2^2. \tag{126}$$

**Proof:** Based on the definition of $\mathcal{I}$, we immediately obtain $\mathbb{E}_{\mathcal{I}}\left[\frac{1}{b}\sum_{i\in\mathcal{I}}\mathbf{z}_i\right] = \mathbb{E}_i[\mathbf{z}_i] = \frac{1}{n}\sum_{j=1}^n\mathbf{z}_j$.

Since $\mathbb{E}_{i,j}[\mathbf{z}_i\mathbf{z}_j] = \mathbb{E}_i[\mathbf{z}_i]\mathbb{E}_j[\mathbf{z}_j] = \mathbf{0}$ for $i \neq j$, we have

$$\mathbb{E}\left[\left\|\frac{1}{b}\sum_{i\in\mathcal{I}}\mathbf{z}_i\right\|_2^2\right] = \frac{1}{b^2}\sum_{i\in\mathcal{I}}\mathbb{E}[\|\mathbf{z}_i\|_2^2] = \frac{1}{b}\mathbb{E}_i[\|\mathbf{z}_i\|_2^2] = \frac{1}{bn}\sum_{i=1}^n\|\mathbf{z}_i\|_2^2. \tag{127}$$

The proof is now complete. $\qquad\square$

**Lemma 5** *Let $\{\mathbf{z}_i\}_{i=1}^n$ be a sequence of $n$ vectors. Let $\mathcal{I}$ be a uniform random mini-batch of $[n]$ with size $b$ (no replacement in samples). Then*

$$\mathbb{E}_{\mathcal{I}}\left[\frac{1}{b}\sum_{i\in\mathcal{I}}\mathbf{z}_i\right] = \frac{1}{n}\sum_{j=1}^n\mathbf{z}_j. \tag{128}$$

*When $\sum_{j=1}^n\mathbf{z}_j = \mathbf{0}$, then*

$$\mathbb{E}_{\mathcal{I}}\left[\left\|\frac{1}{b}\sum_{i\in\mathcal{I}}\mathbf{z}_i\right\|_2^2\right] \leq \frac{\mathcal{I}(b<n)}{bn}\sum_{i=1}^n\|\mathbf{z}_i\|_2^2, \tag{129}$$

*where $I$ is an indicator function, which is equal to $1$ if $b < n$ and $0$ if $b = n$.*

**Proof:** See [23, Lemma A.1]. $\qquad\square$

**Lemma 6** *For variables $\{\mathbf{z}_i\}_{i=1}^n$, we have*

$$\left\|\sum_{i=1}^n\mathbf{z}_i\right\|_2^2 \leq n\sum_{i=1}^n\|\mathbf{z}_i\|_2^2. \tag{130}$$

**Proof**: Since $\phi(\mathbf{x}) = \|\mathbf{x}\|_2^2$ is convex, the Jensen's inequality yields $\|\frac{1}{n}\sum_i\mathbf{z}_i\|_2^2 \leq \frac{1}{n}\sum_i\|\mathbf{z}_i\|_2^2$. $\quad\square$

**Lemma 7** *if $f$ is $L$-smooth, then for any $\mathbf{x}, \mathbf{y} \in \mathbb{R}^d$*

$$|f(\mathbf{x}) - f(\mathbf{y}) - \langle\nabla f_i(\mathbf{y}), \mathbf{x}-\mathbf{y}\rangle| \leq \frac{L}{2}\|\mathbf{x}-\mathbf{y}\|_2^2. \tag{131}$$

**Proof**: This is a direct consequence of A2 [23]. $\qquad\square$

## A.10 Application: black-box classification

**Real dataset** Our dataset consists of $N = 1000$ crystalline materials/compounds, each of which corresponds to a numerical valued feature vector $\mathbf{a}_i$. The feature vector encodes chemical information regarding constituent elements. There exist $d = 145$ attributes, such as, stoichiometric properties, elemental statistics, electronic structure properties attributes, and ionic compound attributes [38]. The label information $y_i \in \{0, 1\}$ (conductor against insulator) is determined using DFT calculations [34]. We equally divided the data into a training and test set.

**Parameter setting**   In our ZO algorithms, unless specified otherwise, the length of each epoch is set by $m = 50$, the mini-batch size is $b = 10$, the number of random direction samples is $q = 10$, the initial value is given by $\tilde{\mathbf{x}}_0 = \mathbf{0}$, and the smoothing parameter follows $\mu = 1/\sqrt{dT}$. For ZO-SGD, ZO-SVRC and ZO-SVRG, we choose $\eta = \frac{5}{d}$ suggested by Corollary 1 and [24, Corollary 3.3]. Also ZO-SVRC updates $J = 1$ coordinates per iteration within an epoch.

## A.11   Application: generating universal adversarial perturbations from black-box DNNs

**Problem formulation**   In image classification, adversarial examples refer to carefully crafted perturbations such that, when added to the natural images, are visually imperceptible but will lead the target model to misclassify. When testing the robustness of a deployed black-box DNN (e.g., an online image classification service), the model parameters are hidden and acquiring its gradient is inadmissible. But one has access to the input-output correspondence of the target model $F(\cdot)$, rendering generating adversarial examples a ZO optimization problem.

We consider the task of generating a universal perturbation to a batch of $n = 10$ images via iteratively querying the target DNN. These images are selected from the class of digit "1" and are all originally correctly classified by the DNN. In problem (1), let $f_i(\mathbf{x}) = c \cdot \max\{F_{y_i}(0.5 \cdot \tanh(\tanh^{-1} 2\mathbf{a}_i + \mathbf{x})) - \max_{j \neq y_i} F_j(0.5 \cdot \tanh(\tanh^{-1} 2\mathbf{a}_i + \mathbf{x})), 0\} + \|0.5 \cdot \tanh(\tanh^{-1} 2\mathbf{a}_i + \mathbf{x}) - \mathbf{a}_i\|_2^2$ be the designed attack loss function of the $i$th image [3, 35]. Here $(\mathbf{a}_i, y_i)$ denotes the pair of the $i$th natural image $\mathbf{a}_i \in [-0.5, 0.5]^d$ and its original class label $y_i$. The function $F(\mathbf{z}) = [F_1(\mathbf{z}), \ldots, F_K(\mathbf{z})]$ outputs the model prediction scores (e.g., log-probabilities) of the input $\mathbf{z}$ in all $K$ image classes. The $\tanh$ operation ensures the generated adversarial example $0.5 \cdot \tanh(\tanh^{-1} 2\mathbf{a}_i + \mathbf{x})$ still lies in the valid image space $[-0.5, 0.5]^d$. The regularization parameter $c$ trades off adversarial success and the $\ell_2$ distortion of adversarial examples. In our experiment, we set $c = 1$ and use the log-probability as the model output. The reported $\ell_2$ distortion is the least averaged distortion over the $n$ successful adversarial images relative to the original images among the $S$ iterations. And in our algorithms, we set $\mu = \frac{1}{\sqrt{dT}}$ and $\eta = \frac{30}{d}$.

**Figure A1:** Black-box attack loss versus number of queries.

**Generated adversarial images**   Table A1 displays the original images and their adversarial examples generated by ZO-SGD and ZO-SVRG. Their statistics are given in Fig. 3. Table A2 shows another visual comparison chart of digit class "4".

**Table A1:** Comparison of generated adversarial examples from a black-box DNN on MNIST: digit class "1".

| Image ID | 2 | 5 | 14 | 29 | 31 | 37 | 39 | 40 | 46 | 57 |
|---|---|---|---|---|---|---|---|---|---|---|
| Original | | | | | | | | | | |
| ZO-SGD | | | | | | | | | | |
| Classified as | 7 | 7 | 3 | 3 | 3 | 3 | 3 | 3 | 3 | 7 |
| ZO-SVRG q = 10 | | | | | | | | | | |
| Classified as | 3 | 3 | 3 | 3 | 3 | 3 | 3 | 3 | 3 | 3 |
| ZO-SVRG q = 20 | | | | | | | | | | |
| Classified as | 7 | 7 | 3 | 3 | 3 | 3 | 3 | 3 | 3 | 7 |
| ZO-SVRG q = 30 | | | | | | | | | | |
| Classified as | 7 | 7 | 3 | 3 | 3 | 3 | 3 | 3 | 3 | 7 |

**Table A2:** Comparison of generated adversarial examples from a black-box DNN on MNIST: digit class "4".

| Image ID | 4 | 6 | 19 | 24 | 27 | 33 | 42 | 48 | 49 | 56 |
|---|---|---|---|---|---|---|---|---|---|---|
| Original | | | | | | | | | | |
| ZO-SGD | | | | | | | | | | |
| Classified as | 9 | 9 | 9 | 9 | 9 | 2 | 9 | 9 | 9 | 9 |
| ZO-SVRG q = 10 | | | | | | | | | | |
| Classified as | 9 | 9 | 7 | 9 | 9 | 5 | 9 | 9 | 9 | 9 |
| ZO-SVRG q = 20 | | | | | | | | | | |
| Classified as | 9 | 9 | 9 | 9 | 9 | 5 | 9 | 9 | 9 | 9 |
| ZO-SVRG q = 30 | | | | | | | | | | |
| Classified as | 9 | 9 | 7 | 9 | 9 | 0 | 9 | 9 | 9 | 9 |