[Reviews · NeurIPS 2018]

Reviewer 1



In this paper, the authors propose a novel variance reduced zeroth-order method for nonconvex optimization, prove theoretical results for three different gradient estimates and demonstrate the performance of the method on two machine learning tasks. The theoretical results highlight the differences and trade-offs between the gradient estimates, and the numerical results show that these trade-offs (estimate accuracy, convergence rate, iterations and function queries) are actually realized in practice. Overall, the paper is well structured and thought out (both the theoretical and empirical portions) and the results are interesting in my opinion (for both the ML and Optimization communities), and as such I recommend this paper for publication at NIPS. - The paper is very well written and motivated, and is very easy to read. - There are some questions/comments that arise about the numerical results presented in the paper: — How is \mu (finite difference interval) chosen? — How are the step lengths chosen for all the methods? — A comparison with other derivative-free optimization methods (e.g., finite difference quasi-Newton methods and/or model based trust region methods) would be of interest. — The authors should show Black-box Attack loss versus Queries (function evaluations) for the second experiment. — The authors should mention how u is chosen in the experiments. - Other minor comments: — The proposed method has many similarities to ZO-SVRC [26]. The authors should clearly state the differences both algorithmic and theoretical. — The authors mention that “this seemingly minor difference yields an essential difficulty in the analysis of ZO-SVRG”, which of course is due to the biased nature of the gradient approximations. Is it fair to say that this is due to the errors in the gradient estimates. If so, the authors should mention that this is the root of the difficulty. — Caption of Figure 3: versus iterations -> versus epochs — Line 287: Sentence starting with “Compared to ZO-SGD,…” Do the authors mean faster convergence in terms of iterations? — What do the authors mean by “accelerated variants”? Accelerated because of the better gradient approximations? — Do the results of Proposition 2 (and as a consequence all the results of the paper) assume precise arithmetic? If so, this should be mentioned by the authors in the paper. Moreover, these results assume that the random directions u are uniform random variables over the Euclidean ball. Could these results be extended to other distributions? — Is it correct to say that CoordGradEst gradient approximations are stochastic (based on subsample of size b) Central Difference approximations to the gradient? — The authors should mention over what the expectations in the theoretical results are taken. — There is proliferating number of derivative-free optimization methods (e.g., finite difference quasi-Newton methods, direct and pattern search methods and model based trust region methods) in the Optimization literature. The authors should mention some of these methods in the paper, and discuss why these methods are not considered.

Reviewer 2



This paper extensively investigates the advantage of variance reduction for stochastic zeroth-order in non-convex optimization (where one does not have access to the gradient of the objective). They consider three different approaches to zo gradient estimation, including RandGradEst, Avg-RandGradEst, and CoordGradEst. For each method, they propose a convergence guarantee. The main contribution (expressed in Theorem 3) is the derivation of the improved rate O(1/T) for CoordGradEst using variance reduction. The result is novel and can potentially have a high impact on non-convex optimization. Although the authors have done a good work to provide a rich theoretical result, I think that the presentation of the paper can be improved a lot. Here, I’ve listed my concerns regarding results and the presentation: 1. My first concern is the dependency of all convergence result to the number of samples n. We can not hide this factor in the O(.) notation because it can crucially change the total time complexity of algorithms (see the seminal paper “Stochastic Variance Reduction for Nonconvex Optimization” where the number of samples is clearly presented in time complexity). 2. The paper uses existing proof techniques of stochastic non-convex optimization and variance reduction without citing them. For example, the Lyapunov function of Eq. 104 is previously used in “Stochastic Variance Reduction for Nonconvex Optimization”. 3. Although the authors have tried to highlight the difference between the difference between different 0-order gradient estimates, I am not convinced that CoordGradEst is the best approach with such a significant improvement in the convergence. The empirical results (in the experiments section) also show that Avg-RandGradEst outperforms as well as CoordGradEst. I think that the source of the convergence gap is the loose bound of Eq 37. If one can prove that \had{\nabla} f_i is Lipschitz in expectation then a same converge rate O(1/T) can be driven for Avg-RandGradEst and even RandGradEst. 4. The proof technique behind all three theorems is the same. I highly recommend to integrate the similar parts of proof in lemmas and propositions instead of repeating all the derivations. If authors represent the upper-bound of proposition 1 for all three 0-order gradient estimates, then reads can understand better the difference between these three approaches. I think that the last term of upper bound vanishes for CoordGradEst and this the source of the improved rate (still I think that this bound can be improved for RandGradEst). Regarding the bound of Eq 99, I could not reach the same upper bound using Eqs (34)-(36). 5. I recommend to first analyze the effect of variance reduction on the smoothed function f_\mu instead of the original function f. Since 0-order gradient estimate provides an unbiased estimate on f_\mu, this helps to simplify the analysis. Then we can relate f_\mu to f in terms of suboptimality. Furthermore, the objective f_\mu as a smooth version of the main function might be more favorable for non-convex optimization. --------------------------------------------- I have read the author response. They addressed most of my concerns, hence I've updated my score. Regarding the 3rd comment, If they assume that stochastic gradients are Lipschitz, they might be able to achieve better bound in Eq. 37.

Reviewer 3



This paper studies the use of variance reduction technique in zeroth order stochastic optimization. The paper particularly focus on plugging in couple of two point gradient estimator in SVRG, e.g. average random gradient estimator and coordinate gradient estimator. Overall, the results make sense and are interesting. The paper is easy to follow. 1. It is a bit difficult to consume some parameters with a long and complicated equation in those theorems. If they are not critical to understand the theorem, they could be simpler. I like Table 1. 2. If combining random coordinate descent and SVRG, the d times more function queries could be reduced to 1. You can refer to the following paper for combing random coordinate descent and SVRG. Randomized Block Coordinate Descent for Online and Stochastic Optimization Huahua Wang, Arindam Banerjee